# Mechanical compressive forces increase PI3K output signaling in breast and pancreatic cancer cells

Mickaël Di-Luoffo[1,2,3], Céline Schmitter[1,2] ORCID, Emma C Barrere[1,2], Nicole Therville[1,2], Maria Chaouki[1,2], Romina D'Angelo[1,2], Silvia Arcucci[1,2], Benoit Thibault[1,2] ORCID, Morgan Delarue[1,3], Julie Guillermet-Guibert[1,2,3] ORCID

Mechanical stresses, including compression, arise during cancer progression. In solid cancer, especially breast and pancreatic cancers, the rapid tumor growth and the environment remodeling explain their high intensity of compressive forces. However, the sensitivity of compressed cells to targeted therapies remains poorly known. In breast and pancreatic cancer cells, pharmacological PI3K inactivation decreased cell number and induced apoptosis. These effects were accentuated when we applied 2D compression forces in mechanically responsive cells. Compression selectively induced the overexpression of PI3K isoforms and PI3K/AKT pathway activation. Furthermore, transcriptional effects of PI3K inhibition and compression converged to control the expression of an autophagy regulator, GABARAP, whose level was inversely associated with PI3K inhibitor sensitivity under compression. Compression alone blocked autophagy flux in all tested cells, whereas inactivation of basal PI3K activity restored autophagy flux only in mechanically non-responsive compressed cells. This study provides direct evidence for the role of the PI3K/AKT pathway in compression-induced mechanotransduction. PI3K inhibition promotes apoptosis or autophagy, explaining PI3K importance to control cancer cell survival under compression.

## Introduction

In tissues, all cells are subjected to mechanical stresses, which correspond to the force per unit surface exerting onto the cell surface (measured in $N/m^2$ or Pascal, Pa), that can notably be transmitted to the nucleus (Lomakin et al, 2020). Physically, the applied force can be either normal or tangential to the surface, causing the cell to deform according to its material properties. Cells encounter three types of mechanical stresses: shear, tensile, and compressive stress (Northcott et al, 2018). These mechanical interactions emerge from cell–cell or cell–substrate interaction (Levental et al, 2009). In cancers, compressive stress is poorly studied and its impact on cell proliferation and migration could depend on the magnitude, duration, and direction of applied forces, and the association with extra-cellular tissue components (Northcott et al, 2018). At homeostasis, epithelial cells sense compressive forces that limit their proliferation (Li et al, 2021); hence, the values of compressive forces applied to epithelial cells are expected to be low compared with the pathological condition. During solid tumor development, tumor cells proliferate rapidly, and this situation is associated with an increase in mechanical forces and an increase in internal rigidity (Therville et al, 2019). In tumors, internal rigidity varies from 10 to 50 kPa (Therville et al, 2019); however, the magnitude of interstitial pressure close to tumor cells is of the order of the hundreds of Pa (Provenzano et al, 2012). Rigidity of the stroma in which the tumor cells proliferate is increasing, promoting tumor cell confinement (Therville et al, 2019). Those events lead to the emergence of large compressive stresses applied to confined tumor cells. In vitro 3D models of solid cancer cell growth under confinement show a cell proliferation decrease (Delarue et al, 2014), whereas colon cancer cell compression without confinement promotes cancer cell growth (Mary et al, 2022). Moreover, compression can increase cancer cell invasive capabilities (Kalli et al, 2019a, 2022). Confinement induces compression stress and increases cell resistance to chemotherapeutic treatments; the latter is notably due to maintained cell survival and reduced proliferation, limiting the effect of both cytotoxic/cytostatic agents (Delarue et al, 2014; Rizzuti et al, 2020). Currently, there is no way to predict what will be the cellular physiological response of compression in cancer cells.

Once exerted on a cell, a mechanical stress induces a mechanotransduction response, coupled with modification of gene expression, which is largely associated with Hippo pathway activation in response to tensile stress (Cobbaut et al, 2020). In tumors, the Hippo pathway containing transcriptional regulators YAP/TAZ can reprogram cancer cells into cancer stem cells and incite tumor initiation, progression, and metastasis (Cobbaut et al, 2020). Furthermore, the Hippo pathway crosstalks with morphogenetic signals, such as Wnt growth factors, and is also regulated by Rho and G protein–coupled receptor (GPCR), cAMP, and PKA pathways (Piccolo

[1]CRCT, Université de Toulouse, Inserm, CNRS, Université Toulouse III-Paul Sabatier, Centre de Recherches en Cancérologie de Toulouse, Toulouse, France  [2]Labex Toucan, Toulouse, France  [3]LAAS-CNRS, Université de Toulouse, CNRS, Toulouse, France

Correspondence: mickael.di-luoffo@inserm.fr; julie.guillermet@inserm.fr

et al, 2014). In the last decade, research has highlighted the interconnections of signaling pathways, and other key intracellular signals involved in mechanotransduction were identified (Di-Luoffo et al, 2021; Schmitter et al, 2023). In this context, the pivotal role of phosphoinositide 3-kinases (PI3Ks) in mechanotransduction of cancers is emerging (Zhao et al, 2018; Borreguero-Munoz et al, 2019). However, PI3K pathway regulation under compressive stress response is not completely elucidated.

PI3K proteins can be divided into three classes (I–III) (Vanhaesebroeck et al, 2010). In the literature, the term PI3K refers usually to class I PI3K, which consists in one p110 catalytic subunit and one p85 regulatory subunit except for PI3Kγ, which harbored p87/p101 regulatory subunits. This class, composed of four enzymes ($α$, $β$, $δ$, and $γ$), with non-redundant functions (Vanhaesebroeck et al, 2010; Arcucci et al, 2021) and non-redundant roles in cancer (Pons-Tostivint et al, 2017), generates phosphatidylinositol 3,4,5-trisphosphate (PtdIns-3,4,5-P3 or PIP3) from phosphatidylinositol 4,5-bisphosphate (PtdIns-4,5-P2 or PIP2) (Vanhaesebroeck et al, 2010). In pancreatic cancer cells, compression-induced PI3K activation promotes migratory phenotype involving an autocrine autostimulation loop (Kalli et al, 2019a, 2022; Thibault et al, 2021). Class I PI3Ks are upstream activators of the YAP/TAZ transcriptional pathway under tensile stress, positioning class I PI3K proteins as potential regulators of essential mechanotransduction signaling (Zhao et al, 2018). In breast cancer cells, the in vivo overexpression of PI3Kβ sensitizes untransformed cells to YAP/TAZ-induced oncogenicity (Zhao et al, 2018). Besides, PI3K/AKT signaling closely regulates autophagy process (Heras-Sandoval et al, 2014; Xu et al, 2020). Furthermore, autophagy process was known to be induced by mechanical shear and tensile stresses in cells in pathological conditions, and in the integration of physical constraints (King et al, 2011; King, 2012; Claude-Taupin et al, 2021). Importantly, the PI3K pathway is one of the most common aberrantly activated pathways in cancers. Genetic alterations of PIK3CA (encoding PI3Kα) lead to constitutively active PI3K/AKT pathway; constitutive PI3K/AKT activation is associated with poor prognosis in pancreatic cancer (Thibault et al, 2021). Cancer dependency to PI3K activation (including breast and pancreatic cancers) provided the rationale for development of inhibitors targeting the PI3K/AKT pathway (reviewed in Ellis and Ma [2019]) for their treatment.

Although the implication of PI3K under tensile stress is now recognized as affecting tumor cell fate (Stylianopoulos, 2017), its role under compressive stress still remains elusive. In particular, the mechanotransduction after a compressive stress remains poorly characterized. Recent evidence showed that among all pro-tumoral signaling pathways, class I PI3Ks appear to be critically involved in the adaptive response to compression (Kalli et al, 2019a; Nam et al, 2019) (reviewed in Di-Luoffo et al [2021]). Hence, PI3K targeting small molecule inhibitors might be relevant therapies to annihilate this adaptive response and lead to cancer cell death. To test this hypothesis, we thus determined the importance of class I PI3K activity in cancer cell lines known to be sensitive to PI3K inhibitors, and using in silico transcriptome analysis and mRNA/protein validation, we identified GABARAP as a common molecular determinant associated with autophagy controlled by compression forces. Breast and pancreatic cancer cells were studied here because mammary and pancreatic solid cancers are commonly subjected to mechanical stress during their development. The purpose of this study is to provide a proof-of-concept showing that the therapeutic impact of targeted therapies is dependent on compressive mechanical context.

# Results

## PI3K pathway inhibition decreases cell number and increases apoptotic cell death in breast and pancreatic cancer cells

First, breast (MCF-7 and MDA-MB-231) and pancreatic (CAPAN-1 and PANC-1) cancer cell lines were treated using a class I pan-PI3K inhibitor (GDC-0941 [10 μM]) or vehicle (Fig 1A).

Interestingly, 72-h treatment with GDC-0941 (10 μM) significantly decreased the crystal violet staining indicative of cell number (cell number for short) in both breast and pancreatic cell lines compared with vehicle (CTL) (Figs 1A and S1), which shows these cells as sensitive to the pan-PI3K inhibitor. These results were confirmed in two additional breast MDA-MB-468 and pancreatic Mia-Paca-2 cell lines (Fig S2A). To investigate the cell death mechanism involved in cell number decrease after PI3K pharmacological treatment, we performed Annexin-V/propidium iodide (PI) cytometry experiments (Figs 1B and S2). The basal cell death status was different in each cell line. Interestingly, GDC-0941 significantly increased early or late apoptosis in pancreatic cancer cells (PANC-1, CAPAN-1; Figs 1B and S3) or in breast cancer cells (MCF-7 and MDA-MB-231; Figs 1B and S2), respectively.

These data corroborate that class I PI3Ks (PI3Kα, PI3Kβ, PI3Kδ, and PI3Kγ) are necessary for breast (MCF-7 and MDA-MB-231) and pancreatic (PANC-1 and CAPAN-1) cancer cell survival.

## PI3K pathway inhibition decreases cell number and increases apoptotic cell death under compression in mechanosensitive breast and pancreatic cancer cells

To investigate the impact of the PI3K/AKT pathway in mechanotransduction after compressive stress, we developed a compression device to compress cells in 2D (Fig 2). The breast and pancreatic cells were compressed with a calibrated 200 Pa pressure per cell adjusting the total mass with small metal weights on an agarose pad (Fig 2). The controls were performed by adding the equivalent volume of the agarose pad medium. We applied both pharmacological inhibition (GDC-0941 [10 μM]) and a 200 Pa 2D compressive force at a single-cell level. Next, we quantified the cell number after 0-, 24-, 48-, and 72-h compression. Interestingly, 200 Pa compression associated with pan-PI3K inhibitor (200 Pa + GDC-0941) significantly decreased cell number in MCF-7 and CAPAN-1, now identified as mechanically responsive cells, whereas GDC-0941 did not affect cell number in mechanically non-responsive cells (MDA-MB-231 and PANC-1) (Fig 3A). These results were confirmed in an additional mechanically responsive Mia-Paca-2 cell line and a mechanically non-responsive MDA-MB-468 cell line (Fig S2B).

Moreover, 200 Pa compression associated with pan-PI3K inhibitor (200 Pa + GDC-0941) significantly increased late apoptosis in MCF-7 breast cancer cells and early apoptosis in CAPAN-1

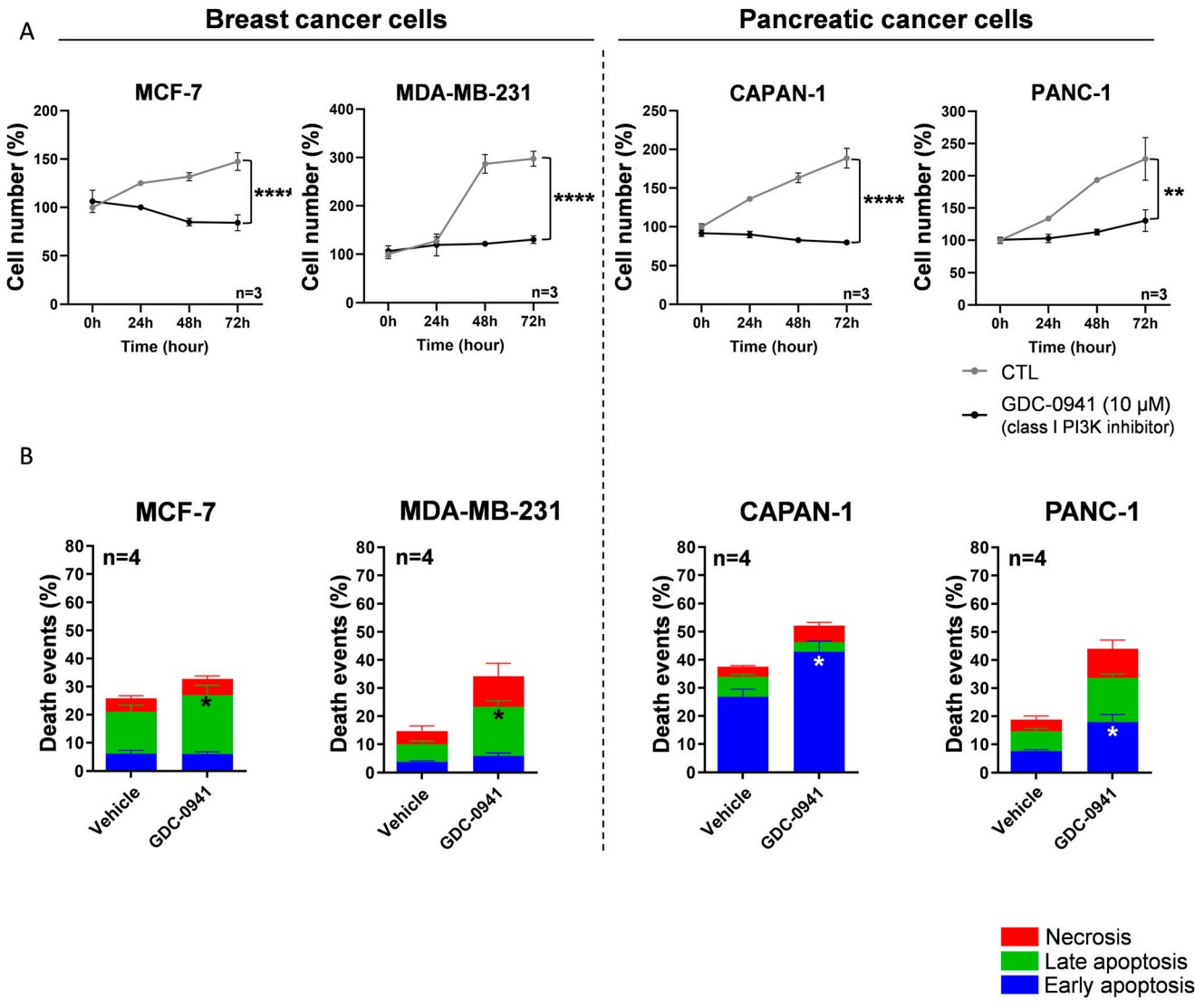

**Figure 1. Cell number and death events in MCF-7/MDA-MB-231 breast and CAPAN-1/PANC-1 pancreatic cancer cells after inhibition of class I PI3K.**
**(A, B)** After cell treatment, the normalized cell number was quantified using crystal violet staining (A, B). **(A)** Cell number was analyzed at 0, 24, 48, and 72 h in MCF-7, MDA-MB-231, CAPAN-1, and PANC-1 cells, treated with class I pan-PI3K inhibitor (GDC-0941 [10 μM]) compared with vehicle. **(B)** After 24-h pan-PI3K inhibitor treatment, cells were stained with a FITC-Annexin-V/PI apoptosis detection kit. FITC-Annexin staining and propidium iodide (PI) incorporation were measured in cells using a flow cytometer and analyzed using FlowJo software. Early/late apoptotic and necrotic cell populations were measured in MCF-7, MDA-MB-231, CAPAN-1, and PANC-1 cells after class I pan-PI3K inhibitor (GDC-0941 [10 μM]) treatment compared with vehicle. Detailed results of living cells, early/late apoptosis, and necrosis are presented in Fig S4. nu, normalized unit. Results are presented as the mean +/− SEM, n = 4. *P*-value after two-way ANOVA, **P* < 0.05.

pancreatic cells that are mechanically responsive cells. However, 200 Pa + GDC-0941 did not affect apoptosis process in mechanically non-responsive cells (MDA-MB-231 and PANC-1) (Fig 3B).

These data support that the cell number decrease in mechanically responsive cells comes from an increase in early and late apoptosis process, which was not affected in mechanically non-responsive cells.

### Compression activates the PI3K pathway in mechanically responsive cells

To experimentally validate the compression impact on the PI3K pathway, we next compressed mechanically responsive cells (MCF-7

and CAPAN-1) and mechanically non-responsive cells (MDA-MB-231 and PANC-1). 200 Pa compression significantly increased by 3.5-fold and 2.7-fold (*P* = 0.04) p-AKT$^{S473}$/total AKT ratio in MCF-7 and CAPAN-1 cells, indicative of PI3K activation in mechanically responsive cells. Furthermore, we analyzed the gene expression of the four class I PI3K genes (*PIK3CA*, *PIK3CB*, *PIK3CD*, and *PIK3CG*). MCF-7 cells harbored a 2.9-fold and 6.2-fold significant overexpression of *PIK3CA* and *PIK3CD*, and CAPAN-1 cells harbored a 2.7-fold (*P* = 0.04), 1.5-fold (*P* = 0.02), and 4.9-fold (*P* = 0.03) significant overexpression of *PIK3CB*, *PIK3CD*, and *PIK3CG*, respectively (Fig S4). Alternatively, 200 Pa compression did not significantly modulate p-AKT$^{S473}$/total AKT ratio in mechanically non-responsive cells MDA-MB-231 and PANC-1 (*P* = 0.21 and 0.26, respectively) (Fig 4). Only

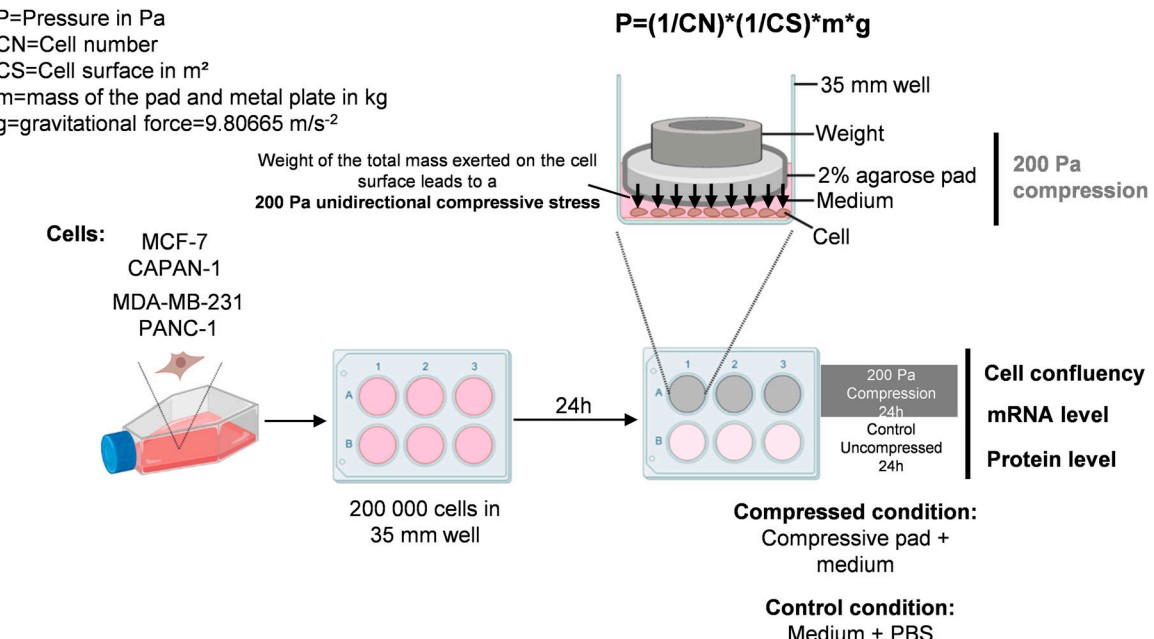

**Figure 2. 2D compression protocol applied in MCF-7, CAPAN-1, MDA-MB-231, and PANC-1 cancer cells.**
MCF-7, CAPAN-1, MDA-MB-231, and PANC-1 cells were plated in a 35-mm petri dish. 24 h after plating, a 2% low gelling temperature agarose pad in PBS was deposited on cells. 2D compression was calibrated at 200 Pa per cell adjusting the mass with a small weight on a 2% agarose pad and calculated using the formula: P = (1/CN)*(1/CS) *m*g (with P, pressure in Pa; CN, cell number; CS, cell surface in $m^2$; m, mass of the pad and metal plate in kg; and g, gravitational force = 9.80665 m/s$^{-2}$). Controls were performed by adding the volume of PBS equivalent to the volume of an agarose pad. After 24 h of compression, cells were harvested, and cell number was calculated and RNA/protein levels were quantified.

*PIK3CG* expression was 11.7-fold significantly increased in PANC-1 (*P* = 0.02) (Fig S4). These results were confirmed in two additional breast MDA-MB-468 and pancreatic Mia-Paca-2 cell lines (Fig S2C). These experimental results show that the activation of the PI3K pathway as assessed by AKT phosphorylation under 200 Pa compression is mostly increased in mechanically responsive cells (MCF-7 and CAPAN-1) but was not affected in mechanically non-responsive cells (MDA-MB-231 and PANC-1). In addition, *PIK3CA*, *PIK3CB*, *PIK3CD*, and *PIK3CG* mRNA overexpression in mechanically responsive cells confirms the AKT activation involves important PI3K pathway members such as class I PI3Ks (PI3Kα, PI3Kβ, PI3Kδ, and PI3Kγ).

### PI3K-AKT pathway–regulated gene expression is sensitive to compression in breast cancer cells

To dive deeper into the molecular mechanism, we decided to analyze transcriptomic data and to understand how compressive stress altered gene signature of cell signaling pathways, including the PI3K pathway. To this end, we searched for datasets obtained upon compression of cancer cells in the Gene Expression Omnibus (GEO) database, where public sequencing data from published studies are available (Fig 5A). In the GEO database, one transcriptomic dataset obtained under cell compression is currently available. Kim et al (2019) compressed breast cancer cells using compressive pads applied on top of 2D cell layers. In this study, Kim et al (2019) focused on and analyzed the action of compression on stromal cells. They observed that mechanical stress in these cells promoted a specific metabolic gene signature increasing glycolysis

and lactate production (Cairns et al, 2011). Here, we compared the gene expression evolution of the two breast cancer cell lines MDA-MB-231 and MCF-7 that harbor decreased cell number under compression. Both breast cancer cell lines were compressed by a global compression applied on all cells ranging from 0 to 8 kPa, and the gene expression profile was analyzed. We first searched for gene signature enrichment analysis using canonical signatures from gene set enrichment analysis (GSEA) (Fig 5A). The gene signatures in different pathways were normalized into transcripts per million and compared with housekeeping gene expression such as *ACTINB*, *LAMINA1*, and *LAMIN* gene class. The expressions of these housekeeping genes were not affected by increasing compression (Fig S5A). Under increasing compression (0), (0.3–0.8), (1.5–4.0), (7.0–8.0) kPa, PI3K-AKT signature gene expression was significantly overexpressed in MDA-MB-231 cells, suggesting a global overexpression of genes belonging to these pathways, whereas it was found already high in MCF-7 cells (Fig S5B, left panel). Those results are in line with the known genetic status of MCF-7 cells, in which PI3Kα is constitutively active by mutation of its encoding gene (*PIK3CA*$^{E545K}$). However, *PIK3CA*, *PIK3CB*, and *PIK3CD* mRNA expressions increased in MCF-7 cells and remained stable in the MDA-MB-231 cell line (Fig S5B, right panel).

Interestingly, the YAP/TAZ pathway gene signature was not significantly affected by increasing compression in both MCF-7 and MDA-MB-231 cells (Fig S5C, left panel). These transcriptomic results were experimentally confirmed by RT–qPCR analyzing the gene expression of YAP/TAZ pathway members such as *TEAD1* and *c-JUN* and of downstream effectors of the YAP/TAZ pathway such as *CCN1*

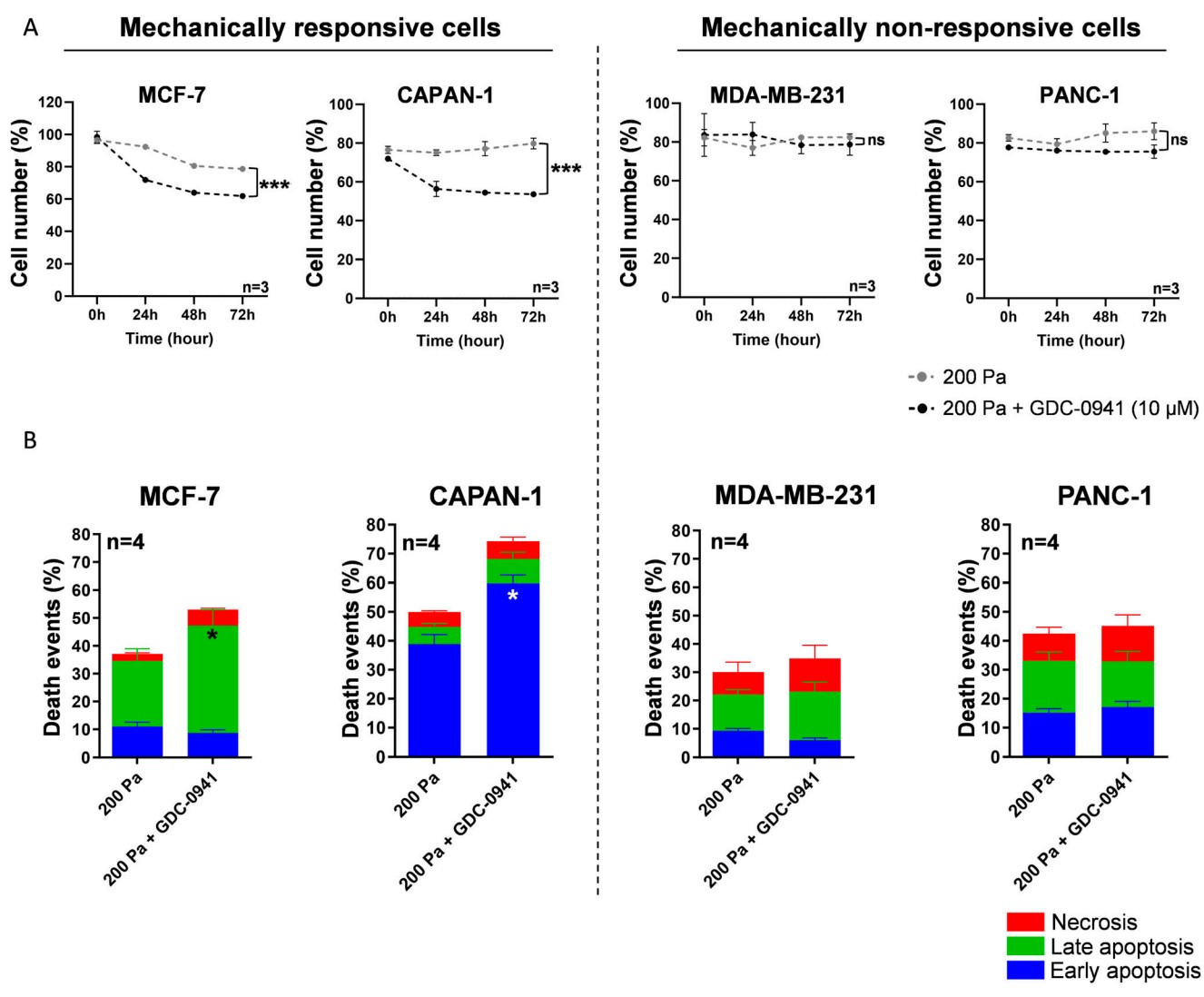

**Figure 3. Cell number and death events in MCF-7/CAPAN-1 mechanically responsive and PANC-1/MDA-MB-231 non-responsive cells associated with inhibition of class I PI3K.**
**(A)** Normalized cell number was analyzed at 0, 24, 48, and 72 h in MCF-7, CAPAN-1, MDA-MB-231, and PANC-1 cells after 200 Pa compression (200 Pa) and class I pan-PI3K inhibitor (200 Pa + GDC-0941 [10 µM]) compared with 200 Pa compressed cells (200 Pa). After cell treatment, the cell number was quantified using crystal violet staining. **(B)** After 24-h pan-PI3K inhibitor treatment, cells were stained with a FITC-Annexin-V/PI apoptosis detection kit. FITC-Annexin staining and propidium iodide (PI) incorporation were measured in cells using a flow cytometer and analyzed using FlowJo software. Early/late apoptotic and necrotic cell populations were measured in MCF-7, CAPAN-1, MDA-MB-231, and PANC-1 cells after 200 Pa compression and class I pan-PI3K inhibitor (200 Pa + GDC-0941 [10 µM]) treatment compared with compressed cells (200 Pa). Detailed results of living cells, early/late apoptosis, and necrosis are presented in Fig S4. nu, normalized unit. Results are presented as the mean +/− SEM, n = 4. $P$-value after two-way ANOVA, $*P < 0.05$.

and *CCN2*. Only the *CCN2* mRNA expression was 1.36-fold significantly increased in MCF-7 mechanically responsive cells (Fig S6A). However, the expression level of other genes was not significantly modulated using 200 Pa compression (200 Pa) in mechanically responsive cells (MCF-7 and CAPAN-1) and in mechanically non-responsive cells (PANC-1 and MDA-MB-231) (Fig S6A). Those transcriptomic data were confirmed by other molecular assays such as the study of YAP phosphorylation that controls its nuclear translocation. The p-YAP$^{S127}$/YAP ratio was not significantly changed under 200 Pa compression in all cell lines (Fig S6B).

Taken together, these data showed an increased gene expression of PI3K family members in cells growing under compression

(from 0 to 8 kPa), especially in mechanically responsive MCF-7 breast cancer cells.

### PI3K pathway–regulated and compression-regulated gene expressions share common targets and control autophagy gene expression in breast cancer cells

To go further using transcriptomic analysis and investigate the specific role of PI3K isoforms in response to compression, we analyzed data from two more studies, which used selective inhibitors of PI3K isoforms: Bosch et al (2015) and Lynch et al (2017). The authors used BYL719 or AZD8186 compounds described as

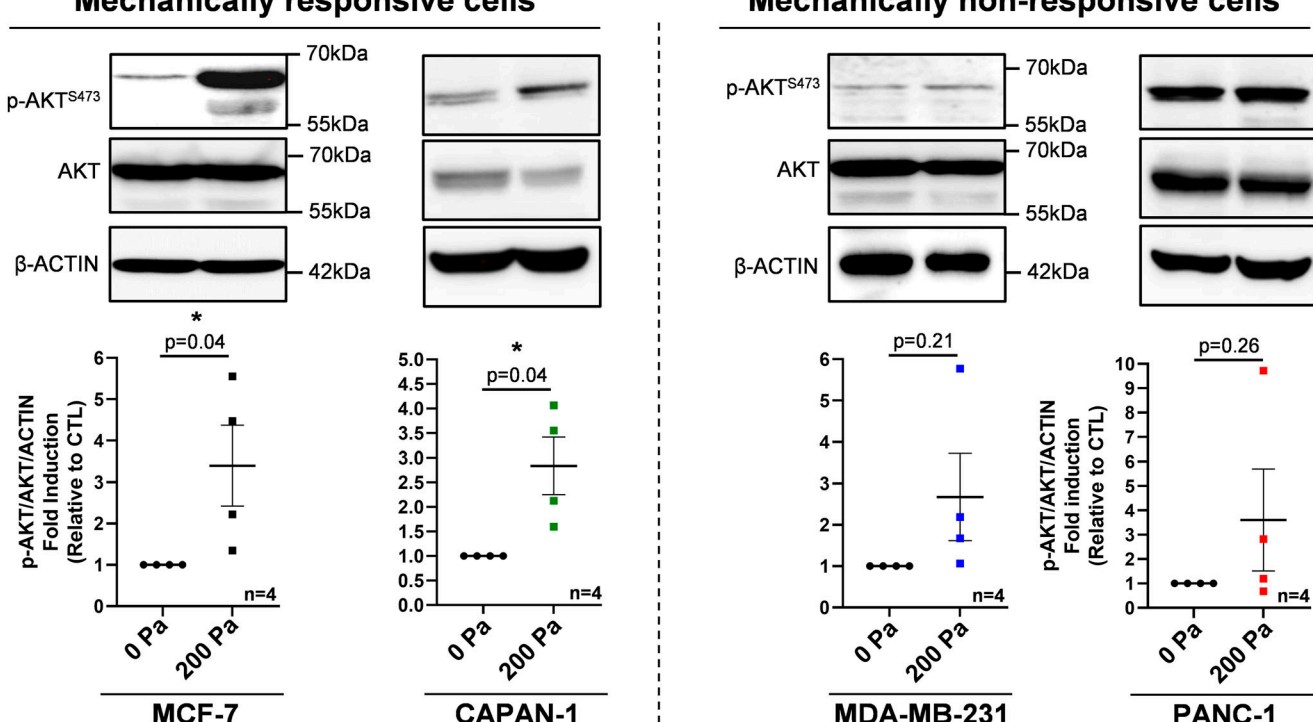

**Figure 4. PI3K signaling in MCF-7/CAPAN-1 mechanically responsive and PANC-1/MDA-MB-231 non-responsive cells.**
Representative Western blots of p-AKT^S473 and AKT in mechanically responsive cells (MCF-7/CAPAN-1) and in mechanically non-responsive cells (PANC-1/MDA-MB-231) after 200 Pa compression (200 Pa) and non-compressed cells (0 Pa) for 24 h. β-Actin was used as a loading control. p-AKT^S473/AKT quantitative analyses were performed using ImageJ software. Results are presented as the mean +/− SEM, n = 4. *P < 0.05. Representation of *PIK3CA*, *PIK3CB*, *PIK3CD*, and *PIK3CG* mRNA expressions in mechanically responsive cells (left panel) and in mechanically non-responsive cells (right panel) after 200 Pa compression is presented in Fig S4.

selective inhibitors of PI3Kα or PI3Kβ, respectively, in mutant *PIK3CA* MCF-7 and mutant *PTEN* HCC70 breast cancer cells. Mutant *PTEN* leads to increased PI3K activity (Chalhoub et al, 2009). We next used gene expression data from PI3Kα inhibition in MCF-7 and PI3Kβ inhibition in HCC70 cells to define PI3Kα and PI3Kβ signatures. Selective transcriptional targets were crossed compared with the compression-specific transcriptional targets that we previously identified (Fig 5A).

After PI3Kα inhibition in MCF-7, 1,279 genes had a significantly altered expression. After PI3Kβ inhibition in HCC70 cells, the gene expression of 933 genes was also significantly affected. The gene expression of 1,052 targets was affected in compressive conditions ([0.3–0.8], [1.5–4.0], and [7.0–8.0] kPa compared with [0] kPa) in MCF-7 cells. We compared these signatures with the list of the 102 genes found in canonical "PI3K-AKT REACTOME signaling pathway in cancer" signature (Fig 5B). PI3Kα and PI3Kβ signatures overlapped with the regulation of 139 genes. Compressive stress gene signature mostly overlapped either with PI3Kα or with PI3Kβ signatures, on non-common 32 and 31 genes, suggesting a differential effect on isoform activation in response to compression.

Finally, GABA type A receptor–associated protein like 1 gene (*GABARAPL1*), coding for GABARAP structural protein of the autophagosome, was the only shared target gene between PI3Kα signature, PI3Kβ signature, compressive stress signature, and "PI3K-AKT REACTOME signaling pathway in cancer" signature (Fig 5B).

*GABARAPL1* and autophagy GSEA pathway gene signatures were affected under increasing compression in breast cancer cells (Figs 5B and S5C). In detail, *GABARAPL1* gene expression was significantly up-regulated after PI3Kα (x1.90; P = 0.08) and PI3Kβ (x1.42; P = 0.006) inhibitions. Under increasing compression from (0) kPa up to (7.0–8.0) kPa, *GABARAPL1* gene expression was significantly increased in mechanically non-responsive MDA-MB-231 cells, but not in mechanically responsive MCF-7 cells (Fig 5B, right panel). Moreover, under increasing compression, the canonical autophagy GSEA pathway gene expression significantly increased in MDA-MB-231 cells and decreased in constitutive PI3K-activated MCF-7 cells (Fig S5C, right panel). These analyses suggest an association between PI3K activation and an increase in autophagy process involved in the non-response of cells to mechanical compressive forces. Moreover, the *GABARAPL1* gene expression alterations converge toward the regulation of autophagosome structural genes.

## Compression modulates the expression of autophagosome structural proteins in breast and pancreatic cancer cells

Autophagy process was proposed as a therapeutic target for various tumors (Xu et al, 2020); inhibition of autophagy in cancer cells decreases their survival. PI3Kα and PI3Kβ are known to be positive and negative regulators of autophagy process (Heras-Sandoval et al, 2014; Xu et al, 2020). When comparing PI3Kα and PI3Kβ

A

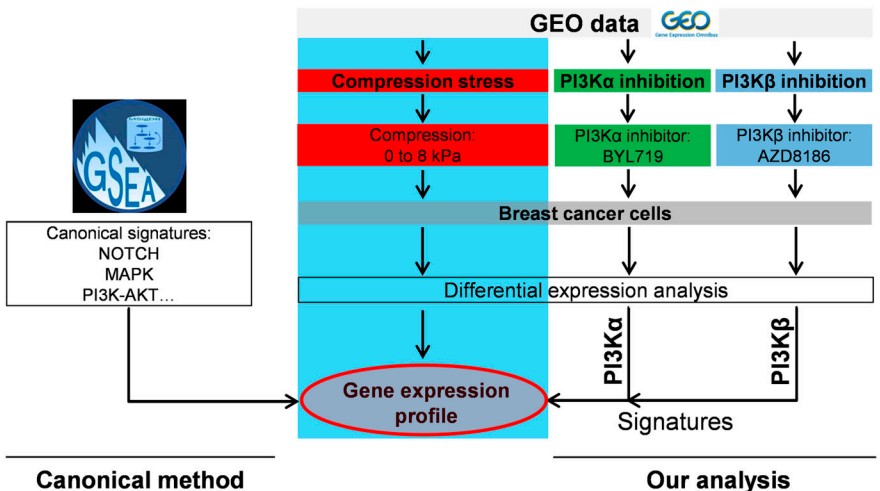

B

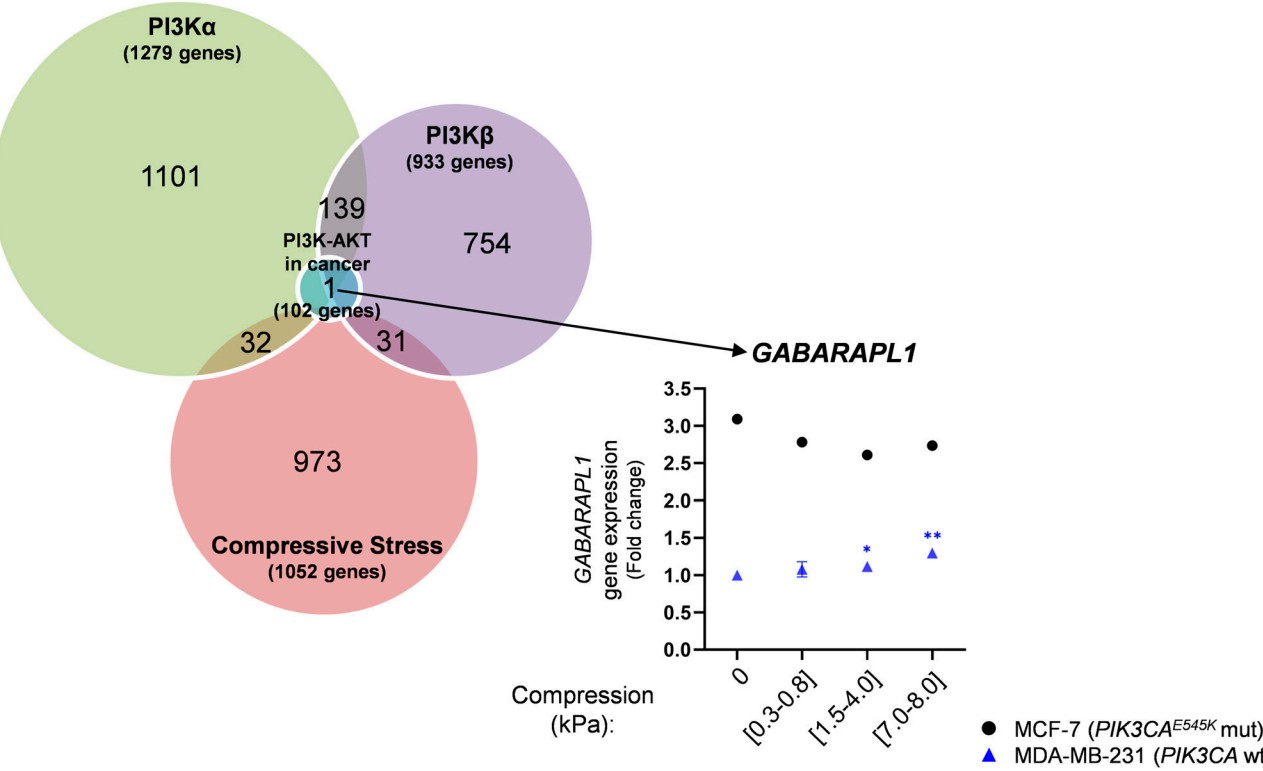

**Figure 5. Comparative analysis between compressive stress, PI3Kα/PI3Kβ, and PI3K-AKT signaling in cancer gene signatures.**
**(A)** Workflow analysis comparing the lists of 1,279, 933, and 1,052 differentially expressed genes in PI3Kα signature, PI3Kβ signature, and compressive stress signature, and 102 genes of REACTOME PI3K-AKT signaling in cancer signature. **(B)** Venn diagram shows the overlapping between PI3Kα signature, PI3Kβ signature, compressive stress, and REACTOME PI3K-AKT signaling in cancer. Overlapping between compressive stress/PI3Kα signature (32 genes), compressive stress/PI3Kβ signature (31 genes), and PI3Kα/PI3Kβ signatures (139 genes) is shown. The only common gene differentially expressed in PI3Kα inhibition/PI3Kβinhibition/compressive stress was *GABARAPL1*. MDA-MB-231 (*PIK3CA*$^{WT}$, blue triangles) and MCF-7 (*PIK3CA*$^{E545K}$ mutated, black dots) breast cancer cells were gradually compressed (0 kPa; [0.3–0.8 kPa]; [1.5–4.0 kPa]; [7.0–8.0 kPa], respectively), and *GABARAPL1* gene expression was quantified using Agilent microarray (data available in https://www.ncbi.nlm.nih.gov/geo/query/acc.cgi?acc=GSE133134).

# Autophagy pathway

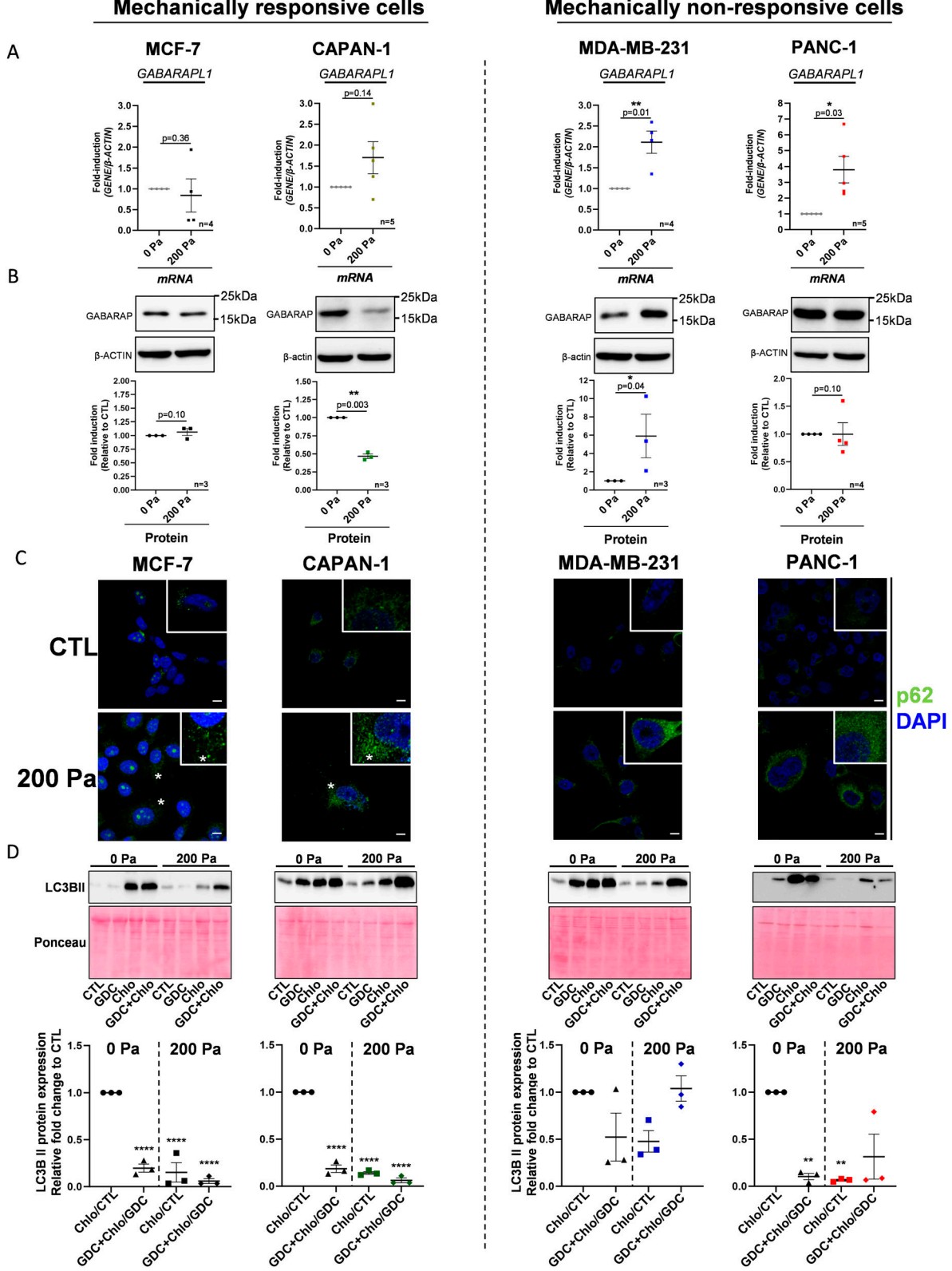

signatures, PI3K-AKT REACTOME canonical pathway, and compressive stress selective gene expression, only *GABARAPL1* gene was significantly affected and overlapped in these four signatures (Fig 5). Briefly, this gene encodes for a structural protein of the autophagosome and autophagolysosome involved in autophagy process (reviewed in Le Grand et al [2011]). To validate transcriptomic data, we decided to analyze autophagy and *GABARAPL1* gene expressions in compressed cells. *GABARAPL1* gene expression was significantly increased in mechanically non-responsive cells (Fig 6). More precisely, *GABARAPL1* gene expression was increased by 2.1-fold (*P* = 0.01) in MDA-MB-231 according to transcriptomic data analysis and by 3.9-fold (*P* = 0.03) in PANC-1 cells (Fig 6A, right panel). These results were confirmed with the GABARAP protein level, which was 5.9-fold significantly increased in MDA-MB-231 (*P* = 0.04) and remains high between uncompressed (0 Pa) and compressed (200 Pa) PANC-1 cells (Fig 6B, right panel). In contrast, GABARAP mRNA and protein levels were not significantly modulated in mechanically responsive cells, except for the decrease in the GABARAP protein level in CAPAN-1 cells in 200 Pa compression condition (Fig 6A and B, left panel), suggesting a decrease of autophagy process in mechanically sensible cells. Furthermore, in the mechanically responsive Mia-Paca-2 cell line, compression did not induce GABARAP protein overexpression. However, in the mechanically non-responsive MDA-MB-468 cells, 200 Pa compression prompted a GABARAP protein increased level (Fig S2B).

These experimental results show that the GABARAP level remains low or decreases in mechanically responsive cells. However, the GABARAP level increases or remains high in mechanically non-responsive cells after compression. The functional link between mechanical insensitivity, autophagy flux, and inactivation of the PI3K pathway therefore will be studied in the remainder of this study.

### Pharmaceutical inhibition of PI3K decreases the autophagy flux only in mechanically responsive cells

Then, we decided to investigate the cellular localization of the autophagosome cargo p62/SQSTM1 by immunofluorescence. Its accumulation in puncta is usually observed when autophagy process is blocked (Bjorkoy et al, 2009). We observed the p62/SQSTM1 level and localization in breast and pancreatic cancer cells under compression. A 200 Pa compression increased the density and the perinuclear localization of p62/SQSTM1 puncta in four cell lines (Fig 6C), whereas larger p62 puncta were only observed in mechanoresponsive cells (Fig 6C, left panel).

We next confirmed whether PI3K inactivation in mechanically responsive cells (MCF-7 and CAPAN-1) might further block autophagy process under compression and quantified the level of the autophagy flux. At autophagosome initiation, LC3B-I is lipidated in LC3B-II (Heras-Sandoval et al, 2014; Xu et al, 2020). Next, LC3B-II autophagosomes are degraded by fusion with lysosomes. To assess the level of LC3B-II autophagy flux, we used chloroquine (10 $\mu$M) that inhibits autophagy by impairing autophagosome fusion with lysosomes (Mauthe et al, 2018), to determine the amount of LC3B-II autophagosomes that is at this time point degraded by lysosomes. Comparing the LC3B-II level of chloroquine-treated cells with controls in the different conditions is thus a means to assess changes in LC3B-II–mediated autophagy flux. We next measured the chloroquine-induced LC3B-II levels in each condition (condition+Chlo/condition ratio) and compared them with the control ratio (Chlo/CTL). In mechanically responsive cells, the LC3B-II autophagy flux was decreased in 0 Pa compression, upon PI3K inhibitor treatment, treating the MCF-7 and CAPAN-1 cell lines with pan-PI3K inhibitor (GDC-0941 [10 $\mu$M]) (GDC+Chlo/GDC) (Fig 6D, left panels). Furthermore, the 200 Pa compression condition decreased even more LC3B-II level with or without pan-PI3K inhibitor (Chlo/CTL or GDC+Chlo/GDC) (Fig 6D, left panels). However, in the mechanically non-responsive MDA-MB-231 cells, pan-PI3K inhibitor (GDC+Chlo/GDC) only or associated with 200 Pa compression did not affect significantly LC3B-II autophagy flux (Fig 6D, right panels). In the mechanically non-responsive PANC-1 cells, pan-PI3K inhibitor (GDC+Chlo/GDC) decreased LC3B-II autophagy flux without compression; however, 200 Pa compression + pan-PI3K inhibitor (GDC+Chlo/GDC) did not affect LC3B-II autophagy flux (Fig 6D, right panels). Taken together, these results showed that although compression blocks autophagy, PI3K inactivation restores autophagy flux in mechanically non-responsive compressed cells.

## Discussion

The impact of mechanical compressive forces on biological processes in cancer cells is still underexplored. Here, our major finding is that high intensity and static compression influences growth rate, cell death, and autophagy process in breast and pancreatic cancer cells that are mechanically responsive or non-responsive cells. PI3K inhibition accentuates this effect, opening avenues for compression in combination with PI3K inhibitors as a therapeutic intervention (Fig 7).

---

**Figure 6. Autophagy flux in MCF-7/CAPAN-1 mechanically responsive and PANC-1/MDA-MB-231 non-responsive cells.**
**(A)** *GABARAPL1* mRNA expressions in mechanically responsive cells (MCF-7/CAPAN-1) and in mechanically non-responsive cells (PANC-1/MDA-MB-231) after 200 Pa compression (200 Pa) and non-compressed cells (0 Pa) for 24 h. $\beta$-Actin was used as a housekeeping gene. Results are presented as the mean +/− SEM. n ≥ 4 for each gene. *P* = *P*-value. \**P* < 0.05; \*\**P* < 0.01. **(B)** Representative Western blots of GABARAP in mechanically responsive cells (MCF-7/CAPAN-1) and in mechanically non-responsive cells (PANC-1/MDA-MB-231) after 200 Pa compression (200 Pa) and non-compressed cells (0 Pa) for 24 h. $\beta$-Actin was used as a loading control. GABARAP/$\beta$-actin quantitative analyses were performed using ImageJ software. Results are presented as the mean +/− SEM, n ≥ 3. *P* = *P*-value. \**P* < 0.05; \*\**P* < 0.01. **(C)** p62/SQSTM1 staining in mechanically responsive cells (MCF-7/CAPAN-1) and in mechanically non-responsive cells (PANC-1/MDA-MB-231) after 200 Pa compression (200 Pa) and non-compressed cells (0 Pa) for 24 h. Green fluorescence corresponds to p62/SQSTM1, and nuclear signal corresponds to DAPI. \* represents puncta. The scale bar corresponds to 10 $\mu$m. n = 3. **(D)** Representative Western blots of LC3B-II in mechanically responsive cells (MCF-7/CAPAN-1) and in mechanically non-responsive cells (PANC-1/MDA-MB-231) after 200 Pa compression (200 Pa) and non-compressed cells (0 Pa) for 24 h. The quantification represents the ratio between 10 $\mu$M chloroquine and control conditions (Chlo/CTL) and between 10 $\mu$M GDC-0941 + 10 $\mu$M chloroquine and 10 $\mu$M GDC-0941 (GDC+Chlo/GDC) in 200 Pa compression (200 Pa) and uncompressed (0 Pa) conditions in all cells. Ponceau S was used as a loading control. LC3B-II/Ponceau S quantitative analyses were performed using ImageJ software and normalized to vehicle 0 Pa. Results are presented as the mean +/− SEM, n ≥ 3. *P*-value after two-way ANOVA, \*\**P* < 0.01; \*\*\*\**P* < 0.0001.
Source data are available for this figure.

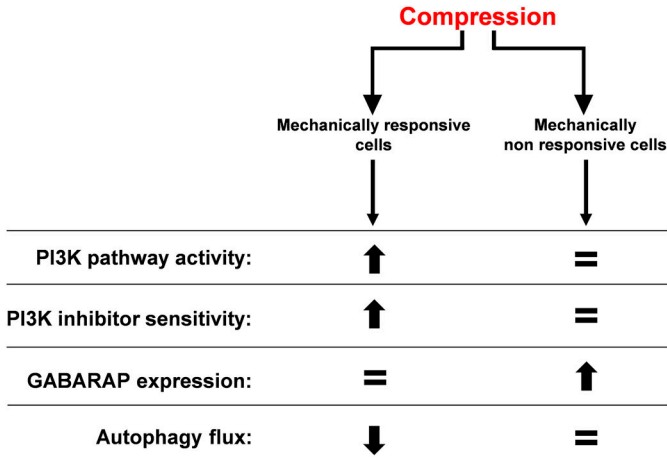

**Figure 7. Balanced impact of compressive stress in mechanically responsive and non-responsive cells.**
PI3K pathway activity, PI3K inhibitor sensitivity, GABARAP expression, and autophagy flux were modulated in mechanically responsive and non-responsive cells. The balance leans toward biological process over/down-activation or actor over/down-expression (black arrows) depending on the cell compressive status.

In terms of molecular mechanisms, the AKT activation by the PI3K pathway has a well-described important role in survival signaling. Activated AKT phosphorylates and inhibits the pro-apoptotic BCL-2 family members BAD, BAX, CASPASE-9, GSK-3, and FOXO1, which tightly regulates apoptosis (reviewed in Miricescu et al [2020]). Hence, besides the pro-migratory (Kalli et al, 2019a) and proliferative (Nam et al, 2019) effects of compression-induced PI3K activation, we show here in breast and pancreatic cancer cells that PI3K can also control cancer cell death processes (early or late apoptosis) under compression.

Through our unbiased approach, we also identify another cell process, autophagy, as critically involved in determining the biological output of compression in cancer cells. Autophagy is a cellular process that allows the orderly degradation and recycling of cellular components, hence providing a self-promoted nutrient source benefiting cell homeostasis and survival (Poillet-Perez & White, 2019). Autophagy is a highly conserved catabolic process, participating in the balance between cell proliferation and cell death in tumor and the tumor microenvironment regulation. This process is highly regulated by the PI3K-AKT-mTOR pathway in various tumors. We know that inhibition of PI3K-AKT-mTOR induces autophagy as part of a tumor growth suppressor role in early cancerogenesis (Yang et al, 2018b; Cui et al, 2019). During cancer initiation, the autophagy flux participates in tissue homeostasis and helps to remove damaged cells. However, autophagy has a dual effect on cancer. In established cancers, autophagy promotes cancer cell survival and resistance to treatment. Autophagy is well known to be a key therapeutic strategy for a variety of tumors including pancreatic cancer (Bryant et al, 2019). Interestingly, PI3K activity controls autophagy in cancer cells with or without compression.

By comparing genes regulated by PI3Kα/PI3Kβ inhibitions, compressive stress, and REACTOME PI3K-AKT signaling in cancer pathway lists of genes, we identified only *GABARAPL1* gene as a shared gene deregulated in these three signatures. Experimental data confirmed the bioinformatics analysis. Compression altered the GABARAP protein level as the GABARAP protein level decreases or remains low in the two cell lines where PI3K inhibition further induces cell death under compression (mechanically responsive cells). The GABARAP protein is known to be cleaved by ATG4B protease before its conjugation with phospholipids. This modified lipidated form is localized on the surface of autophagosomes and lysosomes, participating in their formation. It is therefore involved in autophagy process (reviewed in Le Grand et al [2011]). The differential expression of the GABARAP protein raises the possibility of modulation of autophagy flux (acceleration or blocking) and certainly of the stability of autophagosomes under compressive stress.

There are only limited studies that link induction of autophagy with mechanical stress (Xu et al, 2020; Claude-Taupin et al, 2021). In *Dictyostelium discoideum*, compression activates autophagy in a mTORC1-independent manner (King et al, 2011). This mechanism, if confirmed in mammalian cells, could participate in cancer cell survival and resistance to chemotherapeutic treatments under compression. However, the fact that PI3K inhibition could revert or prevent the blockage of autophagy suggests that mammalian cells might have other mechanisms of cell adaptation to compression.

Under increasing compression, gene/protein expression or phosphorylation status of PI3K-AKT and autophagy pathway members was significantly modulated. However, and surprisingly, "YAP/TAZ via Hippo" or "YAP/TAZ non-Hippo" pathway gene signatures and the expression of YAP/TAZ-regulated genes were not significantly affected in breast and pancreatic cancer cells under unidirectional 2D compressive stress (Fig S6). YAP/TAZ Hippo and non-Hippo pathways are known as key regulators of mechanotransduction under tensile stress (Totaro et al, 2018). In the compressive stress context, the implication of the YAP/TAZ pathway was less investigated. It was assumed that this stress would induce the same signaling pathways as tensile stress. This assumption could have been also prompted by the fact that some techniques used to induce compression could also promote tensile stress (Kalli et al, 2019b; Chen et al, 2019). It is possible that compression in our system could also create a tension on the edge of cell monolayer that is known to be rapidly regulated (Venkova et al, 2022); as we are studying the effect of compression at much higher timescales, the response to compression might prevail in our case. We could argue that a threshold of tensile stress may be needed to permit the activation of YAP/TAZ. If our data seem to confirm those findings in the context of unidirectional compressive stress, differential cytoskeleton remodeling cannot be completely ruled out in our experimental conditions. The exact type of cytoskeleton remodeling under each mechanical stress may also be key to understand cell mechanotransduction processes, as cells adapt to the environmental stresses. Finally, the PI3K signal is permissive for tensile stress control of YAP/TAZ and targeting of PI3K could be a novel strategy to hinder the potential YAP/TAZ oncogenic dependence (Di-Luoffo et al, 2021).

Altogether, we show that breast and pancreatic cancers are sensitive to compression (Kalli et al, 2019a). In tumors, compression forces are mostly a consequence of tumor cell confinement in the

rigid microenvironment that comprises huge ECM accumulation, deposits also called desmoplasia. These high compressive stress situations occur mainly in breast and pancreatic cancers (Verret et al, 2019; Thibault et al, 2021). We previously published that confinement reduced the efficiency of gemcitabine and paclitaxel in pancreatic cancer spheroids because of compression reduced proliferation rate (Rizzuti et al, 2020; Schmitter et al, 2023). Our current study provides an exploratory proof-of-concept of the impact of mechanical stresses on targetable oncogenic cell signaling. The efficiency of both signal-targeted therapies and chemotherapies combined with mechanotherapeutics should, however, be investigated in a wider range of solid cancers, where matrix remodeling is a key component of tumor progression. One widely studied strategy is to target activated PSCs and CAFs, which play an important role in ECM composition, stiffness, and non-autonomous autophagy (Yang et al, 2018a, 2023; Below et al, 2022; Mukhopadhyay et al, 2023). Furthermore, ECM accumulation and desmoplasia may even be heterogeneously distributed in each pancreatic tumor and evolve during progressive stages of the disease (Provenzano et al, 2012); tumoral rigidity is also hetero-geneously distributed (Therville et al, 2019). The importance of this heterogeneity in tumor progression is unknown nowadays. In addition to ECM-induced compressive stress, ECM also influences the tumor progression via epithelial–mesenchymal transition (EMT) and further contributes to resistance to chemotherapeutic agents (Rice et al, 2017).

Others have trialed therapeutic strategies that rely on decompression of solid tumor by reducing surrounding fibrosis and facilitating the accession of chemotherapeutic agents to the tumor (Lampi & Reinhart-King, 2018; Mohammadi & Sahai, 2018; Sheridan, 2019). Some of these trials were stopped because they could accelerate the disease (Mohammadi & Sahai, 2018). As compressive forces decrease proliferation of most cancer cells in in vitro 3D models and here even induce cell death, by reducing the surrounding fibrosis, those therapeutics could allow an increase in cancer cell proliferation and survival and could promote tissue invasion and disease progression. These therapies should thus be considered in combination with conventional chemotherapies that can target newly proliferating cells freed from compressive context. It is even more important that compression of cancer cells without confinement increased cancer cell proliferative behavior (Mary et al, 2022). For these reasons, we are convinced that the balance between therapies reducing fibrosis and fibrosis-induced compression should be well mastered and individually adapted. Our findings also demonstrate that those mechanotherapeutics (modulating the mechanical environment in tumors) could also be used as a way to sensitize to oncogene-targeted therapies such as PI3K inhibitors (Mohammadi & Sahai, 2018).

# Materials and Methods

### PI3Kα genetic knockdown

MCF-7, MDA-MB-231, PANC-1, and CAPAN-1 cells were stably infected using lentiviral transduction containing p110α sh-RNA or scramble-sh-RNA into a pLVTHM plasmid as described in Thibault et al (2021) and listed in Table S1, and selected positively by cell-sorted flow cytometry using GFP expression.

### Cell culture

MCF-7 (#CRL-3435; ATCC), MDA-MB-231 (# CRM-HTB-26; ATCC), MDA-MB-468 (# CRM-HTB-132; ATCC), PANC-1 (#CRL-1469; ATCC), CAPAN-1 (#HTB-79; ATCC), and Mia-Paca-2 (# CRM-CRL-1420; ATCC) cells were cultured in DMEM (#61965026; Gibco) supplemented with 10% FBS (#CVFSVF00-01; Eurobio Scientific), penicillin–streptomycin 50 U/ml and 50 $\mu$g/ml, respectively (#P0781; Sigma-Aldrich), L-glutamine 2 mM (#G7513; Sigma-Aldrich), and Plasmocin 25 $\mu$g/ml (#ant-mpp; InvivoGen) and maintained at 37°C in humidified atmosphere with 5% $CO_2$. Cells were tested for their absence of mycoplasma infection before use. MCF-7 contains constitutive PI3Kα activation ($PIK3CA^{E545K}$) as compared to a MDA-MB-231 cell line that harbors PI3Kα wild-type catalytic domain ($PIK3CA^{WT}$) (Dunn et al, 2019). Both pancreatic cancer cell lines present constitutive PI3K activation (Thibault et al, 2021).

### 2D compression

$2 \times 10^5$ MCF-7, MDA-MB-231, MDA-MB-468, PANC-1, CAPAN-1, or Mia-Paca-2 cells were plated in a 35-mm petri dish. MCF-7 and MDA-MB-231 were plated using DMEM (#61965026; Gibco) supplemented with indicated products in the previous paragraph, however without phenol red. 24 h after plating, a 2% low gelling temperature agarose pad (#A0701-25G; Sigma-Aldrich) in PBS containing 0.5 mM $MgCl_2$ and 0.9 mM $CaCl_2$ respectively (#D8662; Sigma-Aldrich) was deposited on cells. 2D compression was calibrated at 200 Pa per cell adjusting the weight with small metal plates on a 2% agarose pad and calculated using the formula: P=(1/CN)*(1/CS)*m*g (with P, pressure in Pa; CN, cell number; CS, cell surface in $m^2$; m, mass of the pad and metal plate in kg; and g, gravitational force = 9.80665 m/$s^{-2}$). Controls were performed by adding the volume of PBS 0.5 mM $MgCl_2$ and 0.9 mM $CaCl_2$ equivalent to the volume of the agarose pad. Each compression was performed for 24 h, cells were harvested, and RNA/proteins were extracted.

### Normalized cell number assay

Cells were compressed following the protocol described above and/or treated with 10 μM pan-PI3K inhibitor (GDC-0941; Axon Medchem) and/or chloroquine (#C6628; Sigma-Aldrich) for 0, 24, 48, or 72 h. After compression ± treatments, cells were rinsed with PBS (#CS1PBS 0101; Eurobio Scientific) and fixed for 15 min using PBS containing 10% methanol and 10% acetic acid. Cells were stained for 15 min using crystal violet (#HT90132; Sigma-Aldrich). Images were performed using ChemiDoc Imaging System (Bio-Rad) and quantified using ImageJ software.

### Protein extraction

Cells were rinsed with PBS and detached by scrapping in PBS. Cells were harvested through a 5-min centrifugation at 3,000g and 4°C. Cells were resuspended in a lysis buffer containing 150 mM NaCl,

50 mM Tris, 1 mM EDTA, 1% Triton, 2 mM dithiothreitol, 2 mM sodium fluoride, and 4 mM sodium orthovanadate and supplemented with protease inhibitors (cOmplete protease inhibitors, Roche). After a 20-min incubation on ice, a 10-min centrifugation was performed at 12,000$g$ and 4°C and the supernatant was collected. Protein concentration was measured using bicinchoninic acid assay (BC Assay Protein Quantification Kit, #3082; Interchim).

## Western blot

20 $\mu$g of proteins was separated on a 10% polyacrylamide gel and transferred on nitrocellulose membranes (#106000004; Amersham Protran) using Trans-Blot Turbo (Bio-Rad). Blocking was performed through a 1-h incubation in 5% low-fat milk in TBST. Membranes were incubated overnight at 4°C with primary antibodies in TBS/Tween 0.1%, 5% BSA as indicated in Table S2. Then, membranes were rinsed three times with TBS/Tween 0.1%, and incubated for 1 h with secondary antibodies coupled with a horseradish peroxidase, in 1% low-fat milk described in Table S2. Membranes were next washed three times (1 min, 5 min, and 10 min) with TBS/Tween 0.1%. Proteins were detected through chemiluminescence (Clarity Western ECL Substrate; #1705061; Bio-Rad) using ChemiDoc Imaging System (Bio-Rad) and quantified using ImageJ software. ≥3 independent replicates were performed for each protein analyzed.

## Immunostaining

Cells were cultured on glass coverslips. After compression, cells were fixed with 4% PFA in PBS for 10 min and then permeabilized with 0.1% Triton in PBS for 5 min. Cells were blocked in blocking solution (1% BSA in PBS) for 30 min. Samples were incubated with p62/SQSTM1 primary antibody (Table S2) diluted into blocking solution overnight at 4°C. Cells were washed with PBS and then incubated with the Alexa Fluor 488 secondary antibody diluted into blocking solution for 1 h (Table S3). Samples were washed with PBS and incubated with DAPI (#D9542 0.1 $\mu$g/ml; Sigma-Aldrich) as a nuclear counterstain, for 3 min. Coverslips were mounted in Fluoromount-G (# 00-4958-02; Invitrogen).

Images were acquired with a Plan-Apochromat 63x ON 1.4 oil immersion objective using a Zeiss LSM780 confocal microscope.

## Annexin-V-FITC/propidium iodide (PI) assay

2 × 10$^5$ MCF-7, MDA-MB-231, PANC-1, or CAPAN-1 cells were plated in a 35-mm petri dish. 2D compression was calibrated at 200 Pa per cell adjusting the weight with small metal plates on a 2% agarose pad according to the 2D compression section and was treated with 10 $\mu$M GDC-0941 (pan-PI3K inhibitor; Axon Medchem) for 24 h.

After 24 h of compression, cells were washed with fresh PBS, trypsinized (supernatants were kept), and stained with a FITC-Annexin-V/PI apoptosis detection kit (#556547; BD Pharmingen) according to the manufacturer's protocol. FITC-Annexin-V staining and PI incorporation were measured in cells with MACSQuant VYB Flow Cytometer and analyzed using FlowJo software.

## RNA extraction

Cells were rinsed twice with PBS. Cell lysis was performed by scraping cells in one volume of TRIzol reagent (#15596018; Ambion) and incubating for 5 min on ice. Then, 1/5 volume of chloroform (#32211; Sigma-Aldrich) was added to the TRIzol/cell lysate mix. After a 3-min incubation at RT, the cell lysate was centrifuged at 12,000$g$ and 4°C for 10 min, and the supernatant was collected. RNA was precipitated by adding one volume of isopropanol (#33539; Sigma-Aldrich), incubated for 10 min at RT, and then centrifuged at 12,000$g$ and 4°C for 10 min. The RNA pellet was rinsed using 70° ethanol. After a 5-min centrifugation at 7,500$g$ and 4°C, the supernatant was removed, and the RNA pellet was air-dried for 5 min and resuspended in 30 $\mu$l of RNase-free water. RNA concentration was measured using NanoDrop 2000 (Thermo Fisher Scientific).

## Reverse transcription and quantitative PCR

cDNA synthesis was performed with iScript kit (#1708891; Bio-Rad) according to the manufacturer's protocol using C1000 Touch Thermal Cycler (Bio-Rad) and the following conditions: annealing: 5 min at 25°C; reverse transcription: 20 min at 46°C; and reverse transcriptase inactivation: 1 min at 95°C. cDNAs were diluted to a half in RNase-free water. Gene expression was quantified with the SsoFast EvaGreen Supermix kit (#1725204; Bio-Rad) using a thermocycler (#4376374; Software v2.2.2; StepOne) with the following conditions: 20 s at 95°C; 40 denaturation cycles: 3 s at 95°C; and annealing and elongation: 30 s at 60°C. $\beta$-Actin was used as a housekeeping gene. The primers used are described in Table S1, and each amplicon was validated by sequencing. Gene expression quantification was performed using the Livak method: $2^{-(\text{Delta Delta C(T)})}$ (Livak & Schmittgen, 2001). Distinct RNA samples ≥4 were analyzed; each amplification was performed in technical duplicate.

## Transcriptomic analysis

Differential expression analysis under compressive stress in MCF-7 and MDA-MB-231 cells was performed from data provided by Kim et al (2019). MCF-7 and MDA-MB-231 breast cancer cells were gradually compressed (0; 0.3–0.8; 1.5–4.0; and 7.0–8.0 kPa), and gene expression was quantified using Agilent-039494 SurePrint G3 Human GE v2 8x60K Microarray 039381 (data available at https://www.ncbi.nlm.nih.gov/geo/query/acc.cgi?acc=GSE133134). The two PI3K$\alpha$ and PI3K$\beta$ inhibition signatures were performed from differential expression analysis after PI3K$\alpha$ inhibition with BYL-719 (1 $\mu$M for 16–48 h) in MCF-7 breast cancer cells (https://www.ncbi.nlm.nih.gov/geo/query/acc.cgi?acc=GSE64033) (Bosch et al, 2015) or PI3K$\beta$ inhibition after AZD8186 treatment (100 mg/kg twice daily for 5 d) in HCC70 cells (https://www.ebi.ac.uk/arrayexpress/experiments/E-MTAB-4656/) (Lynch et al, 2017). For compressive stress and PI3K$\alpha$ and PI3K$\beta$ inhibition signatures, differential expression analysis was performed using DESeq2 v1.26.0 (Anders & Huber, 2010) on R version 3.6.3 with adjusted $P < 0.05$. All differential expression data are relative data from the normalized transcript per million (TPM). $P$-values on graphs were calculated using a $t$ test comparing no compression condition (0 kPa) with each compression condition ([0.3–0.8], [1.5–4.0], or

[7.0–8.0] kPa). Enrichment analysis was performed using GSEA software, $P < 0.05$, and false discovery rate (FDR) $< 25\%$.

### Statistical analysis

Comparisons between two experimental groups were performed using a paired two-tailed $t$ test. Comparisons between more than two experimental groups were performed using two-way ANOVA. $P < 0.05$ was considered significant. All analyses were performed using GraphPad Prism 9.3.1 software.

# Data Availability

For this work, we used transcriptomic data from compressed breast cancer cells from Kim et al (2019), from the Gene Expression Omnibus (GEO) accession number: GSE133134. Our work correlates compression data to PI3K$\alpha$ inhibition data from Bosch et al (2015) GEO accession number: GSE64033 and PI3K$\beta$ inhibition data from Lynch et al (2017) ArrayExpress: ENA-ERP015852.

# Supplementary Information

# Acknowledgements

We thank our colleagues for their critical reading of the article. C Schmitter is mentored by Ecole Normale Supérieure de Lyon, Université Claude Bernard Lyon 1, France. We thank the Pôle technologique du Centre de Recherche en Cancérologie de Toulouse, for the technical support in imaging and cytometry platforms. Our work on this topic is funded by Fondation Toulouse Cancer Santé (Mecharesist), Inserm Plan Cancer (PressDiagTherapy; MECHAEVO), Inca PLBIO, Toucan ANR Laboratory of Excellence, MSCA-ITN/ETN PIPgen (Project ID: 955534), and Fondation ARC (ARCPJA2021060003932, ARCPGA2022120005630_6362-3). This project has received funding from the European Union's Horizon 2020 research and innovation programme under the Marie Skłodowska-Curie grant agreement No. 955534. C Schmitter and J Guillermet-Guibert obtained a Fondation Fonroga prize for this project.

### Author Contributions

M Di-Luoffo: conceptualization, data curation, formal analysis, supervision, validation, investigation, visualization, methodology, and writing—original draft, review, and editing.
C Schmitter: investigation and methodology.
EC Barrere: investigation.
N Therville: resources, investigation, and methodology.
M Chaouki: formal analysis, investigation, and methodology.
R D'Angelo: validation, investigation, and methodology.
S Arcucci: resources and methodology.
B Thibault: resources, investigation, methodology, and writing—review and editing.
M Delarue: funding acquisition, validation, visualization, project administration, and writing—review and editing.
J Guillermet-Guibert: conceptualization, resources, data curation, formal analysis, supervision, funding acquisition, validation, investigation, visualization, methodology, project administration, and writing—original draft, review, and editing.

### Conflict of Interest Statement

The authors declare that they have no conflict of interest.

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
