## [Reviewer comments · Life Science Alliance]

Life Science Alliance

Mechanical compressive forces increase PI3K output signaling in breast and pancreatic cancer cells

Mickaël Di-Luoffo, Céline SCHMITTER, Emma Barrere, Nicole Therville, Maria Chaouki, Romina D'Angelo, Silvia Arcucci, Benoit Thibault, Morgan Delarue, and Julie Guillermet-Guibert

DOI: <https://doi.org/10.26508/lsa.202402854>

Corresponding author(s): Julie Guillermet-Guibert, Cancer Research Center of Toulouse

Review Timeline:	Submission Date:	2024-05-30
	Editorial Decision:	2024-07-18
	Revision Received:	2024-11-18
	Editorial Decision:	2024-12-13
	Revision Received:	2024-12-17
	Accepted:	2024-12-17

Transaction Report:

July 18, 2024

Re: Life Science Alliance manuscript #LSA-2024-02854

Dr. Julie Guillermet-Guibert
Cancer Research Center of Toulouse
INSERM UMR-1037
2 av Hubert Curien
Toulouse 31037
France

Dear Dr. Guillermet-Guibert,

Thank you for submitting your manuscript entitled "Mechanical compressive forces increase PI3K output signaling in breast and pancreatic cancer cells" to Life Science Alliance. The manuscript was assessed by expert reviewers, whose comments are appended to this letter. We invite you to submit a revised manuscript addressing the Reviewer comments.

Thank you for this interesting contribution to Life Science Alliance. We are looking forward to receiving your revised manuscript.

Sincerely,

B. MANUSCRIPT ORGANIZATION AND FORMATTING:

Reviewer #1 (Comments to the Authors (Required)):

Here authors explore the effect of compressive mechanical forces in solid cancer cells and show evidence of the therapeutic impact in mechanically responsive and non responsive cells. This is certainly an interesting and relevant area of investigation specially in breast and pancreatic cancer which are characterized by an abundant stroma that hinders response to first line therapies.

The manuscript is well written and the results are clearly pointing towards a potential mechanisms where PI3K and GABARAP would play a key role in survival under compressive forces contributing to a better understanding on the mechanobiology of autophagy. Having said that, I found that results and conclusions might not be fully aligned, reason why I would suggest some major revisions:

The authors demonstrate that compression selectively induces overexpression of different PI3K isoforms in mechanically responsive and non responsive cells. This effect leads to a clear activation of PI3K/AKT pathway in the responsive cells under compression (Fig4). However, authors conclude in Fig7 that activation of PI3K is lower under compression in responsive cells. Could authors clarify this point?

Authors state that transcriptional effects of PI3K inhibition and compression converged to control the expression of an autophagy regulator, GABARAP, which level was inversely associated with PI3K inhibitor sensitivity under compression. This appears to be the case in general terms, nonetheless, this conclusion is not evident when looking at results in Fig6. In addition, results in Fig6B shows clear downregulation of GABARAP in CAPAN-1 (responsive cells) and upregulation in MDA-MB-231 (non-responsive) which is not reflected in Fig7.

Due to the disparity of results within the responsive and non-responsive groups, I would suggest to increase the number of cell lines in each group to show more robust data supporting authors' conclusions.

Other minor comments/questions:

In tumors, internal rigidity varies from 10-50 kPa and interstitial pressure close to tumor cells is >100 Pa. Can authors further explain why 200 Pa was applied in the 2D model?

GDC-0941 was used at 10 μ M. I'm wondering why such a high concentration was considered in the experimental design. Wouldn't this inhibitor be effective at the nM range?

Authors have used polymerized agarose pads with a weight to apply compression over the cells. Is this compression uniformly distributed on all cells? Would have been possible to use flexible microfluidic devices in this case?

Since two different cancer cell models are used in this study, it would be interesting to identify common genetic signatures within both responsive cells and compare it with both non responsive.

Authors provide evidence that sensitivity to PI3K inhibitors is dependent on compressive forces. The references cited are relevant and up to date; and the discussion is quite complete including the possibility of using mechanotherapeutics to sensitize cancer cells to oncogene targeted therapies. I probably missed a bit more on the role of activated PSC and CAFs which play an important role on ECM composition, stiffness and non-autonomous autophagy (Yang et al., 2023). What is it known about ECM gene signatures found in co-cultures of tumor organoids and stroma cells? Is there any study on clinically relevant therapies such as KRAS inhibitors, Gemcitabine and Paclitaxel?

Reviewer #2 (Comments to the Authors (Required)):

The authors suggest the importance of PI3K/AKT as a downstream pathway of mechanical compression in regulating cancer cell survival, which is of interest to the cancer research and mechanobiology community. Overall, the article is well-written, and the results are solid and well-presented to support the main points of the paper. I suggest a major revision to address the following comments:

1. Firstly, make sure the definitions of parameters are correct. For example, the definition of stress is incorrect at the second sentence of the introduction. Stress has the unit of Pascal, which is in the same unit as the elastic modulus, i.e. the resistance of sample to deformation. However, stress is not the resistance of samples to deformation.
2. Section 2.2 seems to be a repetition of methods: 2D compression.
3. The authors claim that their method does not implement any tensile stress. However, there is no evidence to support that. I suggest to add in Figure 2 an illustration of how the added weight impose compress to cells may be necessary. This will also help to justify their equation to estimate the compressive stress.

Reviewer #3 (Comments to the Authors (Required)):

In LSA-2024-02854 the authors explore how cancer cells respond to compressive forces in two breast (MCF-7 and MDA-MB-231) and two pancreatic (CAPAN-1 and PANC-1) cancer cell lines. The studies reveal that compressive force can enhance cell death in these cell lines elicited by PI3K inhibition and point toward a mechanistic explanation. Understanding the role of compressive forces on cancer biology is an important, but understudied area. Unfortunately, in the present manuscript how the model of compressive force used in vitro relates to forces experienced by cancer cells in vivo and whether it adequately controls for other variables such that conclusions about the role of compressive forces, per se, are justified, is not adequate for consideration of publication.

Major Issues:

The model used to exert compression on cells embedded in agarose is not supported by previous citations. There are significant concerns about how this models compressive forces in the tumor microenvironment and whether it introduces variables other than compressive force which must be accounted for. Although there is a wide range of variability, externally applied solid stress in tumors is estimated by Jain and colleagues (PMID:25014786) to be 10- to 100-fold higher than the 200Pa model employed here. Moreover, the model involves covering cells embedded in agarose with metal plates for 24h. There is concern here that this disrupts gas/fluid exchange with the embedded cells and induces a hypoxic state which could also influence the biology, independent of compressive force. Other models along these lines employ a trans well insert to allow for exchange (PMID: 31428701). That is not, apparently, the approach here. Also, how uniformly compressive force is distributed to cells in the model is not addressed.

The gene expression dataset analyzed here was generated by another group using a different model, mentioned above, and compression force applies up to 8kPa, significantly higher than that applied in the present manuscript. Thus, how the gene expression analysis relates to the experimental data provided here is unclear. Moreover, there is no functional validation of genes like GABARAPL1 identified in this analysis. Autophagy might also be triggered by nutrient deprivation in this model and how cells respond to that may depend on, among other things, the genetic status of the cell lines. Thus, the data in fig. 6 and conclusions in Fig. 7 may be misinterpreted.

Reviewer 1 (Comments to the Authors (Required)):

Here authors explore the effect of compressive mechanical forces in solid cancer cells and show evidence of the therapeutic impact in mechanically responsive and non responsive cells. This is certainly an interesting and relevant area of investigation specially in breast and pancreatic cancer which are characterized by an abundant stroma that hinders response to first line therapies. The manuscript is well written and the results are clearly pointing towards a potential mechanisms where PI3K and GABARAP would play a key role in survival under compressive forces contributing to a better understanding on the mechanobiology of autophagy. Having said that, I found that results and conclusions might not be fully aligned, reason why I would suggest some major revisions:

♣ The authors demonstrate that compression selectively induces overexpression of different PI3K isoforms in mechanically responsive and non responsive cells. This effect leads to a clear activation of PI3K/AKT pathway in the responsive cells under compression (Fig4). However, authors conclude in Fig7 that activation of PI3K is lower under compression in responsive cells. Could authors clarify this point? **We agree with the reviewer comment: the PI3K activity in responsive compressed cell is higher compared to not compressed cells. The Figure 7 was modified in order to clarify the message according to the reviewer's comment.**

♣ Authors state that transcriptional effects of PI3K inhibition and compression converged to control the expression of an autophagy regulator, GABARAP, which level was inversely associated with PI3K inhibitor sensitivity under compression. This appears to be the case in general terms, nonetheless, this conclusion is not evident when looking at results in Fig6. In addition, results in Fig6B shows clear downregulation of GABARAP in CAPAN-1 (responsive cells) and upregulation in MDA-MB-231 (non-responsive) which is not reflected in Fig7. Due to the disparity of results within the responsive and non-responsive groups, I would suggest to increase the number of cell lines in each group to show more robust data supporting authors' conclusions.

We thank the reviewer for this comment. We confirmed our data in two other cell lines (Mia-Paca-2 and MDA-MB-468) (New Figure S2). Mia-Paca-2 cells were responsive to PI3K inhibition under compression and presented an increased p-AKT/AKT ratio and no change in GABARAP level. MDA-MB-468 cells were non-responsive to PI3K inhibition under compression presenting no increased p-AKT/AKT ratio and an increase in GABARAP level. In conclusion, the addition of two more cell lines confirmed our initial conclusion.

Other minor comments/questions:

♣ In tumors, internal rigidity varies from 10-50 kPa and interstitial pressure close to tumor cells is >100 Pa. Can authors further explain why 200 Pa was applied in the 2D model?

In pancreatic KRAS and p53 mutated cancers, the magnitude of interstitial pressure close to tumor cells was measured in the order of hundred of Pa (Provenzano *et al.*, 2012). Rigidity does not correspond to the mechanical compression pressure. Estimates of accumulated solid stress in the kPa range has been published, as a stored pressure. Moreover, the pressure was measured in 3D. This pressure is also experienced in 2D (unidirectionally). To study the impact on tumor cells of this later stress, here we exert a 2D stress, which could correspond to the stress each cell would experience. As such, it is near impossible to know exactly the local stress cells would experience, but this pressure is expected to range in lower values than rigidity. We have thus undertaken a different approach which would be to apply a minimal stress which would affect cell physiology. The pressure that we applied on each cell was thus calibrated to this range of values using the formula $P=(1/CN)*(1/CS)*m*g$ (with P=Pressure in Pa; CN=Cell number; CS=Cell surface in m²; m=mass of the pad and metal plate in kg; g=gravitational force=9.8 m/s⁻²). Other researchers have also used similar type of calibration: Kalli *et al.*, 2019, applied 2D compression in pancreatic cells between 2 to 4 mmHg = 266 to 533 Pa and Kalli *et al.*, 2021 applied 4 mmHg = 533 Pa pressure on each cell. We have clarified this point in the manuscript.

♣ GDC-0941 was used at 10 uM. I'm wondering why such a high concentration was considered in the experimental design. Wouldn't this inhibitor be effective at the nM range?

We compared the effect of 5 μ M GDC-0941 concentration to 10 μ M. In MCF-7 and PANC-1, 5 μ M GDC-0941 treatment does not affect cell number after 72h exposition while 10 μ M GDC-0941 treatment significantly decreased cell number in these two cell lines. In MDA-MB-231, 5 and 10 μ M GDC-0941 treatment decreased the number of cells in the same way while in CAPAN-1 the cell number decrease was significantly different between 5 and 10 μ M GDC-0941 treatment after 72h exposition (see Reviewer Figure 1 below). To be able to compare effects of treatment across all tested cell lines including in MCF-7 and PANC-1 cells, we chose the 10 μ M concentration that is efficient in all tested cell lines. The data presented in this article are consistent with previously published work: in Thibault *et al.*, 2021, we used 10 μ M GDC-0941 to almost annihilate the pAKT^{S473} levels in the four pancreatic cancer cell lines; high concentrations (10 μ M) were also necessary to reach significant effect on cell number.

Reviewer Figure 1. Cell number in MCF-7/MDA-MB-231 breast and CAPAN-1/PANC-1 pancreatic cancer cells after inhibition of class I PI3K. After cell treatment, normalized cell number was quantified using Crystal Violet staining (A-B). A. Cell number was analyzed at 0, 24, 48 and 72 hours in MCF-7, MDA-MB-231, CAPAN-1 and PANC-1 cells, treated with 5 or 10 μ M class I pan-PI3K inhibitor (GDC-0941) compared to vehicle (CTL). Results are presented as mean, +/- SEM, n=3. *p-value<0.05; **p-value<0.01; ***p-value<0.001.

♣ Authors have used polymerized agarose pads with a weight to apply compression over the cells. Is this compression uniformly distributed on all cells? Would have been possible to use flexible microfluidic devices in this case?

We thank the reviewer for their comment. We chose the agarose pad for being an inert material with little capacity to deformation compared to PDMS (material in which most microfluidic devices are printed) that is deformable.

To confirm that the stress is homogenously distributed, we measured the uniformity of nuclei deformation upon compression at different positions in the well. Each point in the graphs shown in Reviewer Figure 2 (below) is the mean of nucleus areas in one picture. 200 Pa compression significantly increased nucleus areas in 3 cell lines (MCF-7, CAPAN-1 and MDA-MB-231) and tend to increase it in the PANC-1 cell line. Further, the 200 Pa compression affected the nucleus area in the same way throughout the well. As suggested by the reviewer, it would be possible to use microfluidic PDMS devices to apply compression in cells. One potential issue is that the compression imposed by a membrane is hard to calibrate, as the energy inflating the membrane serves both to deform it and to compress cells. Besides, a PDMS membrane would not be permeable to nutrients and would lead to cell starvation, eventually.

Reviewer Figure 2. Nuclei area in MCF-7/MDA-MB-231 breast and CAPAN-1/PANC-1 pancreatic cancer cells after 200 Pa compression. The nuclei area was measured using AxioVert microscope (Zeiss) in four different cell lines and compared between 0 Pa and 200 Pa compression. Each point corresponds to the mean of nuclei areas in one picture done in different fields of the well. p=p-value. Results are presented as mean, +/- SEM. n≥3. *p-value<0.05;**p-value<0.01.

♣ Since two different cancer cell models are used in this study, it would be interesting to identify common genetic signatures within both responsive cells and compare it with both non responsive.

We thank the reviewer for their comment, and it is very interesting to define common genetic signatures. In this study we focused on PI3K and autophagy pathways for which we believe that they are strongly involved in the adaptive response to compression in cancer cells based on previously published work (Di-Luoffo *et al.*, 2021). Future work outside the scope of this manuscript is needed to understand the role of genetic background in the response to compression.

♣ Authors provide evidence that sensitivity to PI3K inhibitors is dependent on compressive forces. The references cited are relevant and up to date; and the discussion is quite complete including the possibility of using mechanotherapeutics to sensitize cancer cells to oncogene targeted therapies. I probably missed a bit more on the role of activated PSC and CAFs which play an important role on ECM composition, stiffness and non-autonomous autophagy (Yang *et al.*, 2023). What is it known about ECM gene signatures found in co-cultures of tumor organoids and stroma cells? Is there any study on clinically relevant therapies such as KRAS inhibitors, Gemcitabine and Paclitaxel? The reviewer comment is totally relevant and appropriate. Few studies associate compressive stress and gene signature (Kim *et al.* 2019; Bosch *et al.*, 2015) and to our knowledge none on ECM gene signatures associated with compressive stress. This topic is a matter of future investigations in our laboratory. We previously published that compression decreased the efficiency of Gemcitabine and Paclitaxel in pancreatic cancer spheroids due to a reduction of proliferation rate (Rizzuti *et al.*, 2020). We are convinced that tumor response to mechanical forces could be part of pancreatic cancer resistance mechanisms to chemotherapies (Reviewed in Schmitter *et al.*, 2023). We added further information on the role of activated PSC and CAFs on ECM composition, stiffness and non-autonomous autophagy in the discussion of the manuscript with references (Yang *et al.*, 2018; Mukhopadhyay *et al.*, 2023; Below *et al.*, 2022; Yang *et al.*, 2023).

Reviewer 2

The authors suggest the importance of PI3K/AKT as a downstream pathway of mechanical compression in regulating cancer cell survival, which is of interest to the cancer research and mechanobiology community. Overall, the article is well-written, and the results are solid and well-presented to support the main points of the paper. I suggest a major revision to address the following comments:

1. Firstly, make sure the definitions of parameters are correct. For example, the definition of stress is incorrect at the second sentence of the introduction. Stress has the unit of Pascal, which is in the same unit as the elastic modulus, i.e. the resistance of sample to deformation. However, stress is not the resistance of samples to deformation.

We thank the reviewer for their comment. The second sentence was modified accordingly, and definition terms were double checked according to the reviewer comment.

2. Section 2.2 seems to be a repetition of methods: 2D compression.

According to the reviewer comment, the paragraph 2.2 was simplified and merged to paragraph 2.3.

3. The authors claim that their method does not implement any tensile stress. However, there is no evidence to support that. I suggest to add in Figure 2 an illustration of how the added weight impose compress to cells may be necessary. This will also help to justify their equation to estimate the compressive stress.

We apologize for the confusion, as we agree with the reviewer. It is possible that compression in our system could also create a tension on the edge of the cell monolayer or to single cells, that is known to be rapidly regulated. As we are studying at higher time scales, the response to compression might prevail, as potential changes in volume due to rapid change in tension are rapidly regulated (Venkova *et al.*, 2022).

Figure 2 and the discussion were modified according to the reviewer comment.

Reviewer 3

In LSA-2024-02854 the authors explore how cancer cells respond to compressive forces in two breast (MCF-7 and MDA-MB-231) and two pancreatic (CAPAN-1 and PANC-1) cancer cell lines. The studies reveal that compressive force can enhance cell death in these cell lines elicited by PI3K inhibition and point toward a mechanistic explanation. Understanding the role of compressive forces on cancer biology is an important, but understudied area.

Unfortunately, in the present manuscript how the model of compressive force used in vitro relates to forces experienced by cancer cells in vivo and whether it adequately controls for other variables such that conclusions about the role of compressive forces, per se, are justified, is not adequate for consideration of publication.

Estimates of accumulated solid stress in vivo in the kPa range has been published, but such stress relates to a stored pressure. Moreover, the pressure is measured in 3D but could be experienced in 2D (unidirectionally) locally. Here we exert a 2D stress, which could correspond to the circumferential stress cells would experience. As such, it is near impossible to know exactly the local stress cells would experience, but this pressure is expected to range in lower values than the stored stress.

We have thus undertaken a different approach which would be to apply a minimal stress which would affect cell physiology. We based our work model on a published methodology (Kalli *et al.*, 2019), that compressed 2D cell monolayers with an agarose pad. We designed a similar methodology to allow comparison and strengthen knowledge acquisition on the topic. Our methodology is indeed simple but allows to decipher the selective role of mechanical compression in cancer biology (Reviewed in Schmitter *et al.*, 2023).

Major Issues:

The model used to exert compression on cells embedded in agarose is not supported by previous citations.

We think there is a misunderstanding. The cells are not “embedded in agarose”, however there is an agarose pad placed over a 2D monolayer of cells. Using this similar method, Kalli *et al.*, demonstrated that PI3K pathway is activated under compressive stress. Cancer cells were grown as a monolayer on the transmembrane and a piston of adjustable weight applied a compressive stress *via* an agarose pad covering the cells.

Supplementary figure 1:

<https://www.ncbi.nlm.nih.gov.proxy.insermbiblio.inist.fr/pmc/articles/PMC6353927/#MOESM1>

There are significant concerns about how this models compressive forces in the tumor microenvironment and whether it introduces variables other than compressive force which must be accounted for.

As discussed above, it is near impossible to know exactly how cells are mechanically challenged in vivo. They could be sheared, pulled or pushed on. In terms of compression, there will certainly be unidirectional compression as well as isotropic compression. In our case, we chose to study the effect of unidirectional compression on tumor cells. The only concern is that it is possible that this pressure could create a tension on cell edge, which opens ion channels. However, such stresses have been shown to be rapidly regulated (Venkova *et al.*, 2022). In our experiments, we work on timescales (days) which are much larger than this specific regulation.

Although there is a wide range of variability, externally applied solid stress in tumors is estimated by Jain and colleagues (PMID:25014786) to be 10- to 100-fold higher than the 200Pa model employed here.

As discussed above, and as pointed out by the reviewer, the stress in a tumor is heterogenous, and what has been estimated by Jain and colleagues is a stored solid stress, which has, among other components, a compressive part. How this compression is locally felt by cells, and what is its magnitude, is near-impossible to measure, and has not been measured, to the best of our knowledge. What we have applied is a minimal stress that has already a large effect on cells.

Moreover, the model involves covering cells embedded in agarose with metal plates for 24h. There is concern here that this disrupts gas/fluid exchange with the embedded cells and induces a hypoxic state which could also influence the biology, independent of compressive force. Other models along these lines employ a trans well insert to allow for exchange (PMID: 31428701). That is not, apparently, the approach here.

Again, cells are not embedded inside the agarose gel, this one is placed over the cells.

To test the hypothesis of the reviewer, we tested hypoxia responses in our compression system. We monitored *HIF1* expression in one mechanically responsive cell line (CAPAN-1) and in one mechanically non responsive cells (MDA-MB-231) by RT-qPCR (primer sequences were taken from van Uden *et al.*, 2008).

Our conclusion was the following: 2D compression of mechanically responsive with our methodology did not modulate *HIF1* expression, and had a little but non-significant effect on non-responsive cells (see Reviewer Figure 3 below). Because hypoxia response was not affected with our method of compression, probably because the pore size of agarose, being in the 100nm range, was not limiting for O₂, we chose not to use transwell system that has also the disadvantage to complicate protein and mRNA extraction.

Reviewer Figure 3. *HIF1* expression in CAPAN-1 mechanically responsive cells and MDA-MB-231 mechanically non-responsive cells after 200 Pa compression. *HIF1* mRNA expression was measured in mechanically responsive cells (CAPAN-1) and in mechanically non-responsive cells (MDA-MB-231) after 200 Pa compression (200 Pa) compared to non-compressed cells (0 Pa) for 24h. β -ACTIN was used as housekeeping gene. Results are presented as mean, +/- SEM. n \geq 4. p=p-value.

Further to the reviewer's comment concerning "metal plates for 24h"; "There is concern here that this disrupts gas/fluid exchange", we tested 2D compression using glass beads instead of metal weight. The

results obtained for cell number in the four cell lines were similar to the metal (stainless steel) weight in control, GDC-0941, 200 Pa and 200 Pa+GDC-0941 conditions (see Reviewer Figure 4 below)

Reviewer Figure 4. Cell number in mechanically responsive cells (MCF-7/CAPAN-1) and mechanically non-responsive cells (MDA-MB-231/PANC-1) after 200 Pa compression. Cell number was quantified at 0, 24, 48 and 72 hours using Crystal Violet staining after 10μM class I pan-PI3K inhibitor treatment (GDC-0941) compared to vehicle (CTL) and 200 Pa compression compared to GDC-0941+200 Pa compression, using glass beads+ agarose pad for compression. n=1.

Also, how uniformly compressive force is distributed to cells in the model is not addressed.

We thank the reviewer for their comment. We chose the agarose pad for being an inert material with little capacity to deformation compared to PDMS (material in which most microfluidic devices are printed) that is deformable.

To confirm that the stress is homogenously distributed, we measured the uniformity of nuclei deformation upon compression at different positions in the well. Each point in the graphs shown in Reviewer Figure 2 (above) is the mean of nucleus areas in one picture. 200 Pa compression significantly increased nucleus areas in 3 cell lines (MCF-7, CAPAN-1 and MDA-MB-231) and tend to increase it in the PANC-1 cell line. Further, the 200 Pa compression affected the nucleus area in the same way throughout the well. As suggested by the reviewer, it would be possible to use microfluidic PDMS devices to apply compression in cells. One potential issue is that the compression imposed by a membrane is hard to calibrate, as the energy inflating the membrane serves both to deform it and to compress cells. Besides, a PDMS membrane would not be permeable to nutrients and would lead to cell starvation, eventually.

The gene expression dataset analyzed here was generated by another group using a different model, mentioned above, and compression force applies up to 8kPa, significantly higher than that applied in the present manuscript. Thus, how the gene expression analysis relates to the experimental data provided here is unclear.

Kim *et al.*, 2019 used a range between 300 Pa to 8 kPa and the autophagy signature in term of gene expression was already significantly increased after 300 to 800 Pa compression in breast cancer cells, confirming the pertinence of our methodology.

Supplementary Figure S5C

Moreover, there is no functional validation of genes like GABARAPL1 identified in this analysis. GABARAPL1 is controlling autophagic flux; both the modulation of mRNA and protein levels of GABARAP in all cell lines was experimentally confirmed. Next, the functional validation that we have already shown in the submitted version is the demonstration that the autophagic flux is differently controlled by compression and PI3K pathway in mechanically-responsive vs mechanically non-responsive cell lines. For clarification, we further highlighted in the manuscript that the functional validation is on the cellular process controlled by compression forces and PI3K signaling. This selective process was identified using unbiased RNAseq data. GABARAPL1 loss-of-function experiments would be a logical next step for future work.

Autophagy might also be triggered by nutrient deprivation in this model and how cells respond to that may depend on, among other things, the genetic status of the cell lines. Thus, the data in fig. 6 and conclusions in Fig. 7 may be misinterpreted.

Agarose pad was chosen because it is permeable to nutrients. In our case, we made it with culture medium. In addition, we showed that the *HIF1* marker was not overexpressed with 200 Pa compression (see above, Reviewer Figure 3), which reflects a non-hypoxic non-nutrient starved state of the compressed cells.

Moreover, in this study, we focus on non-genetic response to compression (oncogenic signaling). Importance of PI3K in this response was also demonstrated using genetic PI3Kα invalidation. We genetically decreased PI3Kα expression using shRNA gene silencing (sh PI3Kα) in the four cell lines. The genetic knockdown decreased for 45 and 44%, the level of the p110α protein (catalytic subunit of PI3Kα) in MCF-7 and CAPAN-1 cells and for 68 and 38% in MDA-MB-231 and PANC-1 respectively (Reviewer Figure 5A-B below). This PI3Kα genetic knockdown decreased p-AKT^{S473} phosphorylation in the four cell lines, indicative of a decreased PI3K-AKT pathway activation (data not shown).

The cell confluency of only mechanically-responsive cells MCF-7 and CAPAN-1 was significantly decreased by combining PI3Kα knockdown (targeting of only one PI3K) and 200 Pa compression compared to Sh Scr / 200 Pa condition (Reviewer Figure 5C left panel). Targeting all PI3Ks using class I pan-PI3K inhibitor (GDC-0941) led to higher effects (Figure 3A). This decrease was not shown in mechanically non-responsive cells (Reviewer Figure 5C right panel).

In conclusion, the MCF-7 and CAPAN-1 have a further compression-induced decrease in cell confluency with PI3K class I pharmacological inhibition or PI3Kα genetic knockdown under compression.

Further work, out of the scope of the manuscript, will be needed to understand the role of genetic background in the response to compression.

Reviewer Figure 5. Cell confluency in mechanically responsive cells (MCF-7/CAPAN-1) and mechanically non-responsive cells (MDA-MB-231/PANC-1) after genetic knockdown of PI3Kα under 200 Pa compression. Representative western blots of p110α in mechanically responsive cells (MCF-7/CAPAN-1) (A) and in mechanically non-responsive cells (MDA-MB-231/PANC-1) (B) cells after genetic knockdown of PI3Kα (sh PI3Kα) vs sh scramble (sh Scr). β-ACTIN was used as loading control. Quantitative analyses were performed using ImageJ software. Results are presented as mean, +/- SEM, n=4. *p-value<0.05, **p-value<0.005. C. Cell confluency was analyzed at 0 and 72 hours in MCF-7, CAPAN-1 and MDA-MB-231, PANC-1 cells after PI3Kα genetic knockdown (sh p110α), 200 Pa compressed or after PI3Kα genetic knockdown (sh p110α) and 200 Pa compressed (dotted curves) compared to control cells. Quantitative analyses were performed using ImageJ software. Results are presented as mean, +/- SEM, n=3. p=p-value; *p-value<0.05; ****p-value<0.0001.

December 13, 2024

RE: Life Science Alliance Manuscript #LSA-2024-02854R

Dr. Julie Guillermet-Guibert
Cancer Research Center of Toulouse
INSERM UMR-1037
2 av Hubert Curien
Toulouse 31037
France

Dear Dr. Guillermet-Guibert,

Thank you for submitting your revised manuscript entitled "Mechanical compressive forces increase PI3K output signaling in breast and pancreatic cancer cells". We would be happy to publish your paper in Life Science Alliance pending final revisions necessary to meet our formatting guidelines.

- please consider Reviewer 3's remaining comments, however I disagree that the manuscript overstates GABARAPL1's role as a functional mediator of the response to compression. Please go through the paper a final time to make sure this is not overstated.
- please be sure that the authorship listing and order is correct
- please upload your main and supplementary figures as single files
- you have a callout for Figure S7, but this figure is not uploaded and there isn't a figure legend for it; please correct

Figure Check:

- please add weights next to all blots

LSA now encourages authors to provide a 30-60 second video where the study is briefly explained. We will use these videos on social media to promote the published paper and the presenting author (for examples, see <https://docs.google.com/document/d/1-UWCfbE4pGcDdcgzcmiuJI2XMBJnxKYeqRvLLrLS08s/edit?usp=sharing>). Corresponding or first-authors are welcome to submit the video. Please submit only one video per manuscript. The video can be emailed to contact@life-science-alliance.org

A. FINAL FILES:

B. MANUSCRIPT ORGANIZATION AND FORMATTING:

Sincerely,

Reviewer #1 (Comments to the Authors (Required)):

Authors have thoroughly reviewed the comments, clarified all the points and added further data confirming their initial hypothesis. In my opinion, authors have appropriately addressed the limitations of the study and concerns regarding the mechanical compression offering opportune references which support the rationale behind their experimental design, observations, and conclusions. I congratulate authors for their work and wish them all the best in their future investigations which will definitely be of high interest for the scientific community.

Reviewer #3 (Comments to the Authors (Required)):

The authors have substantially addressed concerns related to the model and some other issues raised previously. There is still some question about the causal role of GABARAPL1 (or autophagy more generally) as a mediator of the response of cells under compressive force to PI3K inhibitors.

The authors should provide more compelling experimental evidence that GABARAPL1 is a functional mediator of the response to compression-knocking it down should sensitize mechanically non-responsive cells to PI3K inhibitors under compression. Without this they should remove the GABARAPL1 data until it can be validated.

Alternatively, the authors should test the role of autophagy, at least by using a pharmacologic inhibitor of autophagy like chloroquine to test whether that sensitizes mechanically-non-responsive cells to PI3K inhibitors under compression.

Without these experiments the role of autophagy generally and GABBARAP1 specifically remain correlative with regard to their role in mediating the biology of cancer cells under compression.

Reply to reviewers:**Reviewer 1** (Comments to the Authors (Required)):

Authors have thoroughly reviewed the comments, clarified all the points and added further data confirming their initial hypothesis. In my opinion, authors have appropriately addressed the limitations of the study and concerns regarding the mechanical compression offering opportune references which support the rationale behind their experimental design, observations, and conclusions. I congratulate authors for their work and wish them all the best in their future investigations which will definitely be of high interest for the scientific community.

We aim to thank the reviewer for all their important and constructive comments which improved the quality of our study.

Reviewer 3 (Comments to the Authors (Required)):

The authors have substantially addressed concerns related to the model and some other issues raised previously. There is still some question about the causal role of GABARAPL1 (or autophagy more generally) as a mediator of the response of cells under compressive force to PI3K inhibitors. The authors should provide more compelling experimental evidence that GABARAPL1 is a functional mediator of the response to compression-knocking it down should sensitize mechanically non-responsive cells to PI3K inhibitors under compression. Without this they should remove the GABARAPL1 data until it can be validated.

Alternatively, the authors should test the role of autophagy, at least by using a pharmacologic inhibitor of autophagy like chloroquine to test whether that sensitizes mechanically-non-responsive cells to PI3K inhibitors under compression.

Without these experiments the role of autophagy generally and GABBARAP1 specifically remain correlative with regard to their role in mediating the biology of cancer cells under compression.

We have gone through the manuscript to make sure we have not overstated GABARAPL1's role as a functional mediator of the response to compression. We agree with the reviewer that the GABARAPL1 loss-of-function experiments would be a logical next step for future work.

December 17, 2024

RE: Life Science Alliance Manuscript #LSA-2024-02854RR

Dr. Julie Guillermet-Guibert
Cancer Research Center of Toulouse
INSERM UMR-1037
2 av Hubert Curien
Toulouse 31037
France

Dear Dr. Guillermet-Guibert,

Thank you for submitting your Research Article entitled "Mechanical compressive forces increase PI3K output signaling in breast and pancreatic cancer cells". It is a pleasure to let you know that your manuscript is now accepted for publication in Life Science Alliance. Congratulations on this interesting work.

DISTRIBUTION OF MATERIALS:

Again, congratulations on a very nice paper. I hope you found the review process to be constructive and are pleased with how the manuscript was handled editorially. We look forward to future exciting submissions from your lab.

Sincerely,
